

# Development of a multi-fusion convolutional neural network (MF-CNN) for enhanced gastrointestinal disease diagnosis in endoscopy image analysis

Tanzim Hossain[1], F M Javed Mehedi Shamrat[2], Xujuan Zhou[3], Imran Mahmud[1], Md. Sakib Ali Mazumder[1], Sharmin Sharmin[2] and Raj Gururajan[3]

[1] Department of Software Engineering, Daffodil International University, Dhaka, Bangladesh
[2] Department of Computer System and Technology, University of Malaya, Kuala Lumpur, Malaysia
[3] School of Business, University of Southern Queensland, Springfield, Australia

Corresponding authors
F M Javed Mehedi Shamrat,
javedmehedicom@gmail.com
Xujuan Zhou,
xujuan.zhou@usq.edu.au

## ABSTRACT

Gastrointestinal (GI) diseases are prevalent medical conditions that require accurate and timely diagnosis for effective treatment. To address this, we developed the Multi-Fusion Convolutional Neural Network (MF-CNN), a deep learning framework that strategically integrates and adapts elements from six deep learning models, enhancing feature extraction and classification of GI diseases from endoscopic images. The MF-CNN architecture leverages truncated and partially frozen layers from existing models, augmented with novel components such as Auxiliary Fusing Layers (AuxFL), Fusion Residual Block (FuRB), and Alpha Dropouts (αDO) to improve precision and robustness. This design facilitates the precise identification of conditions such as ulcerative colitis, polyps, esophagitis, and healthy colons. Our methodology involved preprocessing endoscopic images sourced from open databases, including KVASIR and ETIS-Larib Polyp DB, using adaptive histogram equalization (AHE) to enhance their quality. The MF-CNN framework supports detailed feature mapping for improved interpretability of the model's internal workings. An ablation study was conducted to validate the contribution of each component, demonstrating that the integration of AuxFL, αDO, and FuRB played a crucial part in reducing overfitting and efficiency saturation and enhancing overall model performance. The MF-CNN demonstrated outstanding performance in terms of efficacy, achieving an accuracy rate of 99.25%. It also excelled in other key performance metrics with a precision of 99.27%, a recall of 99.25%, and an F1-score of 99.25%. These metrics confirmed the model's proficiency in accurate classification and its capability to minimize false positives and negatives across all tested GI disease categories. Furthermore, the AUC values were exceptional, averaging 1.00 for both test and validation sets, indicating perfect discriminative ability. The findings of the P-R curve analysis and confusion matrix further confirmed the robust classification performance of the MF-CNN. This research introduces a technique for medical imaging that can potentially transform diagnostics in gastrointestinal healthcare facilities worldwide.

# INTRODUCTION

The prevalence of gastrointestinal illnesses (GI) is a major public health problem. There are around 2.8 million new cases of GI disorders each year, with an additional 1.8 million deaths attributable to esophageal, colorectal, and stomach malignancies. The Pan American Health Organization (PAHO) reports that there were 375,170 fatalities in the Americas in 2019, with 160,002 of those deaths attributable to digestive issues. The overall crude mortality rate due to GI is 37.2 per 100,000 persons (43.3 per 100,000 males and 31.3 per 100,000 females). From 2000 to 2019, both the overall death toll and the crude death toll *per capita* rose. From top to bottom, the countries with the highest age-standardized mortality rates are Honduras, Guatemala, Bolivia, Haiti, Guyana, Mexico, and Nicaragua (*PAHO, 2021*).

In Bangladesh, 25–40% of the population suffers from gastrointestinal (GI) disorders. Out of a total of 3,000 participants in research (*Perveen, Rahman & Saha, 2014*), 2,273 (75.8%) reported experiencing at least one upper GI symptom in the previous 3 months, while 2,072 (69.1%) reported experiencing two or more symptoms. Additionally, 1,705 (56.8%) reported experiencing three or more symptoms. Upper abdomen discomfort was experienced by 963 participants (32.1%), bloating by 1,265 participants (42.16%), heartburn by 1,354 participants (45.13%), chest pain by 1,166 participants (38.87%), early satiation by 1,347 participants (44.9%), and vomiting by 258 participants (8.6%). Digestive problems are widespread. Multiple gastrointestinal symptoms are possible simultaneously. The prevalence of symptoms is influenced by their intensity, frequency, and duration.

According to medical professionals, anyone who has chronic symptoms, including rectal bleeding, nausea, appetite loss, stomach aches, or other symptoms connected to GI illnesses, should have an endoscopy performed to determine the underlying cause. An endoscope used during an endoscopy allows medical professionals to look for infections or potential cancer indications while detecting uncommon characteristics in the human GI tract. However, most patients see endoscopy as a prolonged and painful therapy due to the endoscope's design and its first admission *via* the mouth. Fortunately, researchers developed the wireless capsule endoscopy (WCE) technology, which enhanced the procedure and resolved the issues above. In contrast to standard endoscopy, which entails putting a lengthy tube-like camera down the patient's throat, the WCE method only employs a small illuminated camera within a typical-sized capsule that the patient eats orally. The capsule, once swallowed, may make its journey gently from the back of the throat all the way up to the small intestines (*Wang et al., 2013*).

The WCE represents a significant advancement in the examination procedure. By offering an enhanced patient experience without the difficulties typically associated with

traditional methods, WCE provides a comprehensive visual of the GI tract. The capsule's design is specifically intended to aid physicians in detecting abnormalities faster and more accurately. The mucosa's outside appearance is used in diagnostics. However, the presence of certain precursors or illnesses, such as polyps, ulcerative colitis, or esophagitis, may indicate a more severe problem. Even with WCE technology, endoscopic evaluation of the disease above remains complex and time-consuming, resulting in incorrect diagnoses. Rapid and exceptionally precise data with enhanced consistency might help physicians make more informed treatment choices. WCE has the benefit of collecting many images of the GI tract, but analyzing these images still requires medical personnel to draw on their clinical expertise, which may be both time-consuming and difficult, not to mention mentally stressful and draining. As a result, there is a higher chance of inaccurate diagnoses or delayed data publication because of human limitations (*Kumar et al., 2017*).

According to recent research, several deep learning methods and computer vision have improved GI endoscopic image diagnosis (*Haile et al., 2022*; *Naz et al., 2021*). Despite the inclusion of a CNN module, many studies still necessitated complex procedures and relied on meticulous manual feature extraction techniques. Researchers used strategies such as conventional methods for deep learning, CNN, CRNN, and OCR of NLP models. Few have attempted to combine pre-trained models. However, only one has predicted or detected gastrointestinal illnesses using this strategy. *Montalbo (2022)* introduced the MFuRe-CNN model to automate GI disease identification from endoscopic data. This model incorporated three advanced deep convolutional neural networks: EfficientNet, MobileNetV2, and ResNetV2. A layer-wise fusion was achieved by truncating, partially freezing, and reconfiguring their layers using AuxFLs with αDOs. After fusing the features, a FuRB with αDO managed the combined data and minimized overfitting. The model obtained 97.75% accuracy on the test dataset and 96.65% on the validation set for four classes. However, it indicated a deficiency in its resilience, had a restricted capacity for benchmarking, and may not exhibit consistent performance across diverse medical images. The effectiveness of the system might have been improved by using more advanced deep convolutional neural networks (DCNNs).

On the other hand, *Fan et al. (2018)* used a modified AlexNet, a traditional DCNN model, to identify intestinal erosions and ulcers from WCE images. The images were preprocessed to eliminate blackened areas, reducing potential false positives. The number of major class neurons in AlexNet was reduced from 1,000 to 2, and two ReLU and dropout layers were added to improve stability. The model obtained a 95.16% accuracy rate for the two classes but faced challenges with resilience due to a limited dataset and shortened network, resulting in some misdiagnoses outside the training data. In contrast, *Majid et al. (2020)* used the VGG16, a renowned DCNN model, to diagnose gastrointestinal disorders. While the original VGG16 was recognized for its performance in the ILSVRC-2014 challenge, *Majid et al. (2020)* enhanced it by incorporating custom feature engineering methods like the discrete wavelet transform and strong color features. These modifications expanded its feature set. After using K-nearest neighbors to combine the finest features, they trained the model with a genetic algorithm, attaining 96.5%

accuracy on a 9,889-image training set. However, the model's complexity poses replication and deployment challenges.

However, *Poudel et al. (2020)* introduced a unique CNN model with dynamic dilated Conv layers, regularized using their DropBlock regularizer, to classify diverse colorectal disorders. They argued that excessive downsampling from numerous pooling layers in extended networks often leads to loss of spatial information. To counter overfitting due to limited data, they incorporated the DropBlock regularizer. The study underscored the significance of appropriate layer adjustments and regularization in enhancing CNN performance. Their method surpassed many DCNNs, obtaining a 95.7% accuracy. However, its computational complexity and high operating costs potentially impact performance. The research highlighted the risk of overfitting and information loss in extensive networks on limited datasets. Besides, *Hmoud Al-Adhaileh et al. (2021)* introduced a robust framework to classify gastrointestinal tract diseases using the Kvasir dataset. They employed three deep learning models: AlexNet, GoogleNet, and ResNet-50. These models processed 9,216 features and directed them to fully connected layers, resulting in 1,000 neurons. The softmax layer then classified each image into one of five gastrointestinal disorder categories. All models demonstrated promising results, but AlexNet had the highest accuracy rate of 97%.

Hence, *Khan et al. (2020)* trained the VGG16 model on a small dataset using transfer learning, fine-tuning, and feature fusion, avoiding complex custom methods. By leveraging pre-existing features from ImageNet and integrating feature fusion into the Cubic Support Vector Machines, they achieved 98.4% accuracy in distinguishing between a GI ailment and a healthy digestive tract. Their approach emphasized the significance of transfer learning in GI disease diagnosis, but it only offered binary classification without addressing multi-class GI disorders. Despite this, *Öztürk & Özkaya (2021)* integrated a Residual-LSTM module into the ResNet50 DCNN model for categorizing endoscopic data. Using residual learning, they reduced the saturation effect often seen in deeper DCNN models. This combined approach led to an accuracy of 98.05%, surpassing the results of AlexNet and GoogleNet when equipped with the same module. The study concludes that residual-based models, like ResNet50, are more effective in diagnosing GI tract diseases than models focusing only on layer depth.

Furthermore, *Montalbo (2022)* introduced a deep learning-based lightweight and cost-efficient state-of-the-art (SOTA) method utilizing KVASIR and ETIS-Larib Polyp DB datasets. The proposed approach seamlessly integrates network compression, layer-wise fusion, and the incorporation of a customized residual layer, denoted as the Modified Residual Block (MResBlock). The regularization process involves the application of a self-normalizing technique, which yielded an impressive accuracy of 96.65% on the validation dataset and further elevated its efficacy to 97.75% on the test dataset in the context of diagnosing four cases of GI tract conditions and surpassing other pre-existing solutions in the domain. Subsequently, *Zhang et al. (2023)* investigated the Swin transformer model and explored two methods, attention block and MoCo pre-training, with an accuracy of 87.22% to categorize the upper gastrointestinal system into 12 classes. The authors developed a gastric navigation system that uses a DL-based image classifier for

identification and keeping track of the area being viewed of the GI system on the esophagogastroduodenoscopy (EGD) video feed.

In addition, *Thomas Abraham et al. (2023)* proposed a strategy for supplementing transfer-learning deep CNN models using the KVASIR dataset to identify digestive diseases. ResNet50, InceptionV3, DenseNet121, and EfficientNetB0 are utilized for this experiment. Among these models, EfficientNetB0 attains the best accuracy of 98% for five diseases (dyed lifted polyps, normal cecum, normal pylorus, polyps, and ulcerative colitis) classification task. Similarly, *Gunasekaran et al. (2023)* proposed a novel approach GIT-Net, which employed pre-trained InceptionV3, ResNet50, and DenseNet201 CNN models in the role of feature extractors for the KVASIR v2 endoscopic image dataset to produce a probability correctness of 95% with the proposed weighted average ensemble method for GI tract classification.

*Mushtaq et al. (2023)* also present a deep learning-based novel framework attention-based SSD for gastric polyps (ASSD-GPNet) model. The strategy employed a single-shot multi-box detector (SSD) combining VGG-16 for feature extraction, and the refined map block (RMB) was incorporated into SSD's High-Res feature maps to attain more semantic information of the utilized dataset 1970 gastric images and Pascal VOC07 + 12. The ASSD-GPNet achieved mean average precision (mAP) score of 0.942 on gastric images and 0.769 on Pascal VOC. In the meantime, for segmentation of the gastrointestinal tract, authors in *Sharma et al. (2023)* suggested an improved U-Net model design employing six transfer learning models InceptionV3, SeResNet50, VGG19, DenseNet121, InceptionResNetV2, and EfficientNetB0 as the backbone of the U-Net topology. The proposed U-net model stands apart with its distinctive design and deviating from the inclusion of dense layers in the model, an integrated combination of convolution, max pool, and transpose convolution layers achieving 0.122 model loss, 0.8854 dice coefficient, and 0.8819 IoU in terms of performance analysis.

Recent developments have provided enormous advances *via* several kinds of different methods in the rapidly developing area of gastrointestinal (GI) disease diagnostics. Despite these developments, there remain issues, particularly with the accuracy that is essential for a reliable disease diagnosis. Diagnostic accurateness is often impacted by the ineffectiveness and limited adaptability of existing techniques such as residual learning, regularization, feature fusion, layer reconstruction, and model compression due to their intricate procedures, intricate features extracted, and scalability issues. Our research presents the Multi-Fusion Convolutional Neural Network (MF-CNN) as a solution to these complex issues. The MF-CNN effectively combines feature fusion, residual learning, and unique regularization techniques. It differs from the conventional transfer learning method by using more extensive layer transformations and fine-tuning procedures to achieve improved accuracy. The MF-CNN offers an important advantage in its simplified and effective workflow, functioning autonomously without the need for additional feature extraction and complex learning techniques. This minimizes the possibility of errors that may result from these auxiliary procedures. The model utilizes Auxiliary Feature Layers (AuxFLs) to augment the layers, α-Dropout (αDOs) for enhanced regularization, and Fusion Residual Blocks (FuRBs) to combine residual learning with feature fusion. These

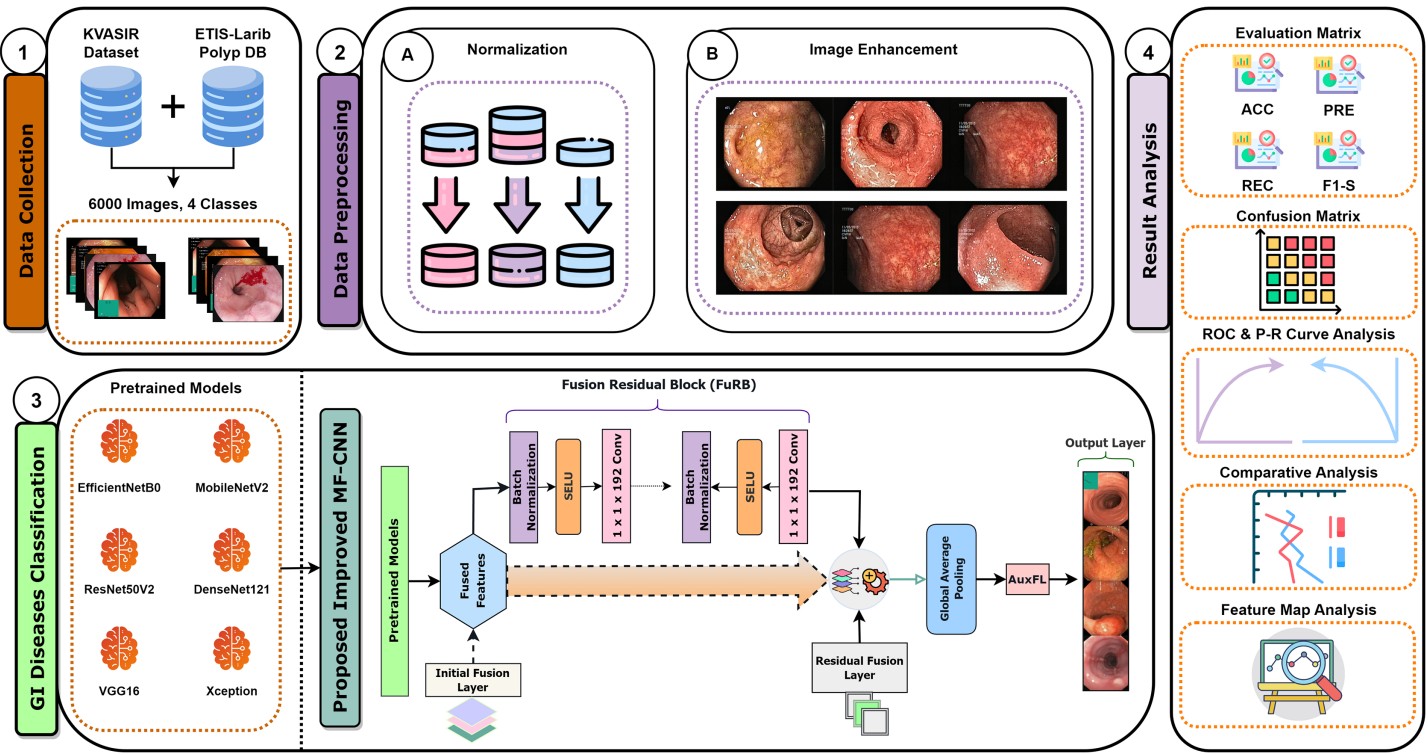

**Figure 1** **MF-CNN integrates six pre-trained models.** AuxFLs, αDOs, and FuRBs highlight layer enhancement and feature fusion, simplifying GI disease diagnosis.

elements are carefully chosen to manage robustly fused features effectively, aiming to enhance performance while simplifying the architecture. This method effectively addresses the difficulties of accuracy and adaptability which were highlighted in the previous study. The main advancement of this study is the integration of six pre-trained models into a single pipeline. This appropriately addresses both issues of accuracy and limitations associated with adaptability in various clinical scenarios. This study aims to improve the diagnosis process and significantly increase the accuracy of identifying gastrointestinal diseases by using deep convolutional neural networks (DCNNs). Figure 1 illustrates the complete workflow of our proposed method. The key contributions of this study are as follows:

- Introduced a Multi-Fusion Convolutional Neural Network (MF-CNN) that integrates Auxiliary Fusing Layers, Fusion Residual Block, and Alpha Dropouts for accurately identifying GI diseases, presenting a more accurate and cost-effective alternative to convolutional models.
- Incorporated AHE in the preprocessing step to enhance the contrast and quality of the endoscopic images, improving the accuracy of disease classification.
- A detailed ablation study was conducted to enhance the robustness and accuracy of the MF-CNN, examining and fine-tuning its components, ensuring optimal performance across various scenarios.
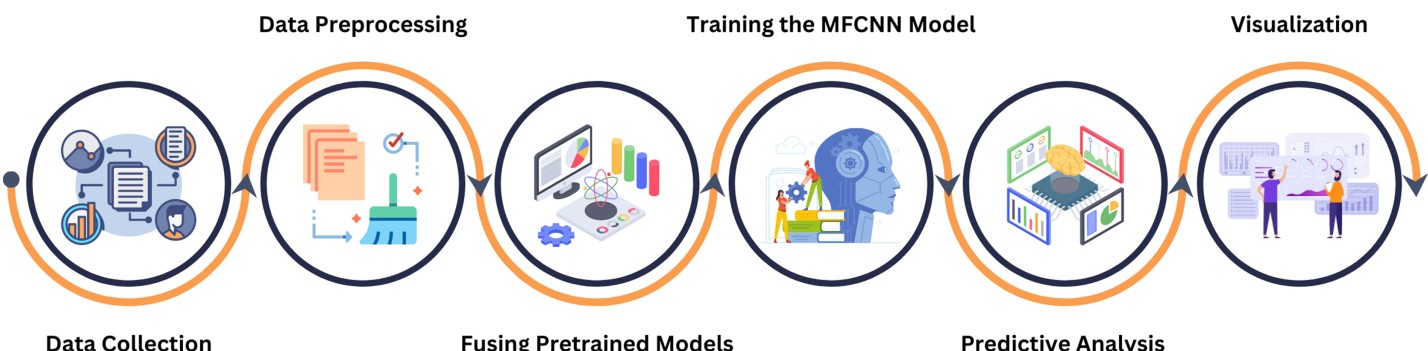

**Figure 2** This figure delineates the various phases undertaken in our systematic approach, providing a comprehensive overview of the methodology from the beginning to ending.

- Employed a comprehensive set of evolution metrics, achieving an impressive accuracy of 99.25%, demonstrating the model's ability to classify and minimize false classifications across all diseases accurately categorized.
- Provided a detailed feature map for understanding and interpreting the model's outputs, ensuring transparency and reliability.

The remainder of the article is organized as follows: The next section, 'Procedural Approach and Techniques', elaborates on dataset collection, preprocessing, and development approach of the proposed MF-CNN model. The following section, 'Ablation Analysis' analyzes the impact of each component on the model's performance. Subsequently, the 'Comparative Analysis of All Applied Models' section covers the experimental setup and the model's computational aspects. The following section 'Result and Analysis Findings' describes the study's experimental findings. Then, the next section 'Limitations of the study', acknowledges the potential constraints and areas for improvement. Finally, the 'Conclusions' section summarizes key findings, contributions, and the impact of our research in medical image analysis and GI tract abnormality detection.

## PROCEDURAL APPROACH AND TECHNIQUES

This study employs wireless capsule endoscopy (WCE) imaging data to detect gastrointestinal diseases. The images in their raw form have been obtained from several databases. For optimum model performance, these raw images go through comprehensive preprocessing to meet the specific demands of the model. The proposed model incorporates elements from various established convolutional neural network architectures, ensuring robustness in its predictions. After training on the collated image data, the model demonstrates remarkable accuracy in diagnosing gastrointestinal diseases. Our research approach is depicted in a structured six-step procedure, as highlighted in Fig. 2.
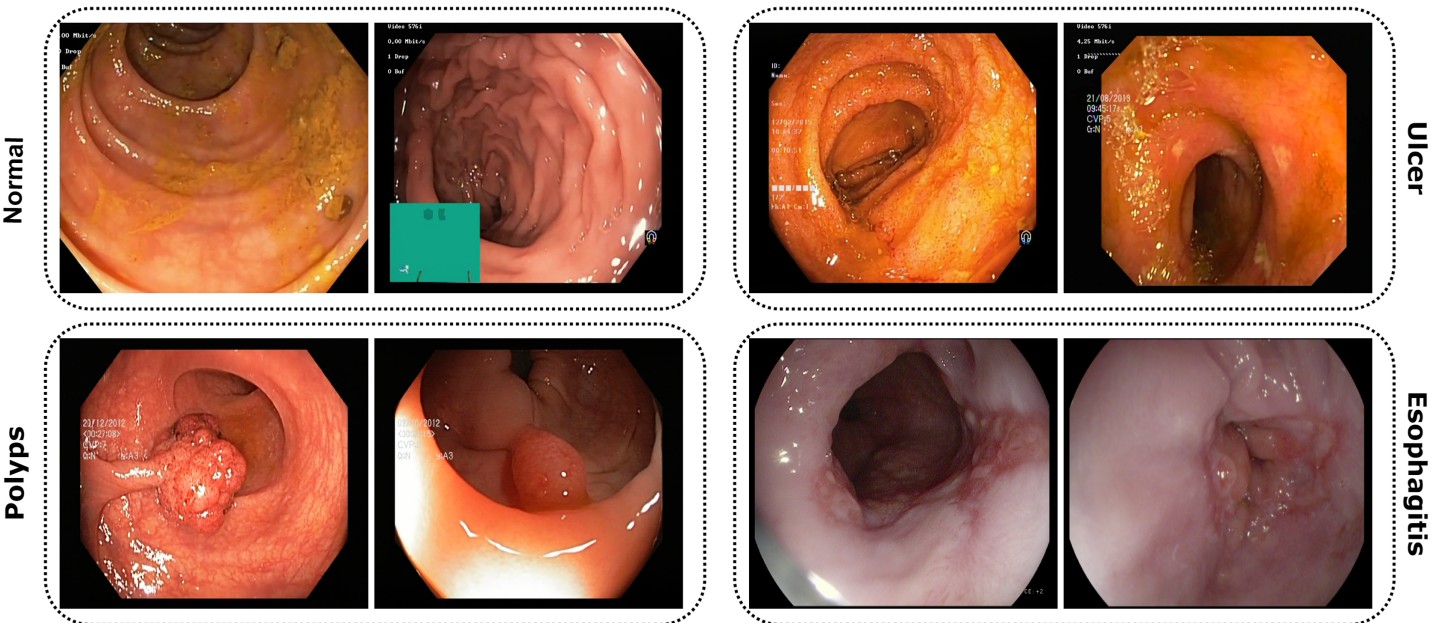

**Figure 3** Images from the dataset show four classes: Normal, Ulcer, Polyps, and Esophagitis. Each image highlights features of its GI condition from capsule endoscopy.

## Data collection

The study utilized publicly accessible datasets from reputable repositories, including KVASIR (*Pogorelov et al., 2017*) and ETIS-Larib Polyp DB (*Silva et al., 2014*). Due to the complexities associated with obtaining endoscopic data related to gastrointestinal disorders, these datasets proved invaluable. The KVASIR dataset was derived using the wireless capsule endoscopy (WCE) technique from a diverse population of gastrointestinal patients at the Vestre Viken Health Trust in Norway. Upon comprehensive review and data acquisition, medical specialists ensured each image within the dataset was correctly labeled, making it suitable for most deep-learning studies. Similarly, the verification process was applied to the ETIS-Larib Polyp DB dataset. In this inquiry, a conscious decision was made to utilize a stochastic approach in order to choose and organize the datasets carefully. This was done to minimize the possibility of data leakage and any potential biases. Figure 3 of the dataset showcases samples from each category.

## Comprehensive data overview

In this study, we employed a meticulously prepared dataset comprising a total of 6,000 images. This dataset was strategically divided into distinct subsets to facilitate a systematic approach in model training and evaluation: training, validation, and testing. The distribution was set as 3,200 images for training, 2,000 for validation, and 800 for testing. Such a structured segmentation assures a proportionate exposure of the model to varied data while also allowing for a comprehensive evaluation of its performance. One of the primary concerns in machine learning, particularly in image classification tasks, is the

**Table 1 Detailed dataset specification and distribution.**

| Class | Train | Validation | Test | Total |
|---|---|---|---|---|
| Normal | 800 | 500 | 200 | 1,500 |
| Ulcer | 800 | 500 | 200 | 1,500 |
| Polyps | 800 | 500 | 200 | 1,500 |
| Esophagitis | 800 | 500 | 200 | 1,500 |
| Total | 3,200 | 2,000 | 800 | 6,000 |

challenge posed by data instability and class dominance. Our methodology adeptly addresses and minimizes these possible drawbacks by assuring an even distribution of images across various subsets. For a detailed dissection of the dataset, including class categorization, quantity, and specific distribution, refer to Table 1. A noteworthy characteristic of the images within the dataset is the variance in their dimensions, attributable to their diverse sources of origin. Recognizing the need for uniformity in input data for deep learning models, we standardized the training images to a consistent dimension of 224 × 224 pixels. This was accomplished using an automatic image data generator available within the Keras framework (*Chollet, 2015*). Such a standardization facilitates the computational process and ensures efficient resource allocation. By adopting this technique, we optimized the training speed while concurrently preventing computational memory overload, ensuring smooth and efficient experimentation.

## Data preprocessing

### Data normalization

Data sourced from diverse origins often exhibit variations in format and structure. Such inconsistencies can potentially compromise the efficacy of deep convolutional neural networks (DCNNs). Specifically, models may encounter challenges in achieving adequate convergence during training if these inconsistencies remain unaddressed (*Swati et al., 2019*). To counteract this issue, our study employed a normalization strategy. All pixels, represented as xi, in each image, were rescaled using the min-max scaling Eq. (1). This procedure ensured that every image had pixel values consistently scaled to a range of 1.0

$$xi = \frac{xi - min(xi)}{max(xi) - min(xi)}. \tag{1}$$

### Image enhancement (adaptive histogram equalization)

In our study, we recognized the potential benefits of enhancing the clarity and contrast of the images to obtain more precise results. To this aim, we employed the adaptive histogram equalization (AHE) technique (*Pizer et al., 1987*), applying specific parameters of 'clip_limit = 2.0' and 'tile_grid_size = (8, 8)' to adjust the process to our needs. AHE is renowned for its capacity to amplify the local contrast of an image, particularly in areas that are closer in color or intensity. The fundamental principle of AHE involves performing histogram equalization within small, contextual regions or tiles of the image

rather than the entire image. For each tile, defined by the 'tile_grid_size' parameter as $8 \times 8$ blocks, the histogram of pixel intensities denoted as $H(i)$, is calculated for each tile. It provides the frequency at which each intensity level $i$ arises.

$$H(i) = number\ of\ pixels\ with\ intensity\ i. \tag{2}$$

Following this, the cumulative distribution function *(CDF)* is computed for each histogram using the formula $C(i)$,

$$C(i) = \sum_{j=0}^{i} H(j). \tag{3}$$

In this equation, $j$ is an index runs from 0 to $i$, summing the histogram values up to the current intensity level $i$, to calculate the cumulate frequency. After calculating the CDF, it is normalized with the 'clip_limit' parameter set to 2.0. This normalization process is represented as $C_{norm}(i)$, which adjusts the intensity values within each tile to enhance local contrast,

$$C_{norm}(i) = Normalized\ Value\ of\ C(i). \tag{4}$$

The notation $C_{norm}(i)$ refers to the value of the normalized *CDF* for the intensity level $i$. Applying AHE to our dataset with these parameters improved the images by increasing the visibility of identifiable features. This improvement facilitated a more comprehensive representation and set an environment for our model to effectively detect and learn from complex, intricate patterns in the image. The effectiveness of AHE in the study is demonstrated in Fig. 4, which clearly displays the enhancement of contrast in local regions of the images. The histograms display the distribution of pixel intensities across the RGB color channels for both the original and AHE-enhanced images, providing a visual representation of the contrast adjustments. This processing phase ensured that our model was trained on data that accurately reflected the depth and clarity of the original endoscopic images, increasing the probability of accurate diagnosis. Algorithm 1 illustrates the entire AHE procedure.

## Development approach

This research explored an extensive variety of deep convolutional neural networks (DCNNs) for their efficacy in medical image analysis. Here, we present a systematic review and integration of various models, leading to the development of the proposed MF-CNN for diagnosing gastrointestinal (GI) conditions.

## Selection and evaluation of DCNN models

Convolutional neural networks (CNNs), a form of artificial neural network, play a crucial role in deep learning, particularly when analyzing visual data (*Hossain et al., 2022*). These networks consist of many connected layers of artificial neurons. Their design focuses on feature extraction and classification, with a composition of an input layer, convolutional layer, pooling layer, fully connected layer, hidden layer, and activation function. The crucial aspect of this study included utilizing the capabilities of DCNNs, known for their

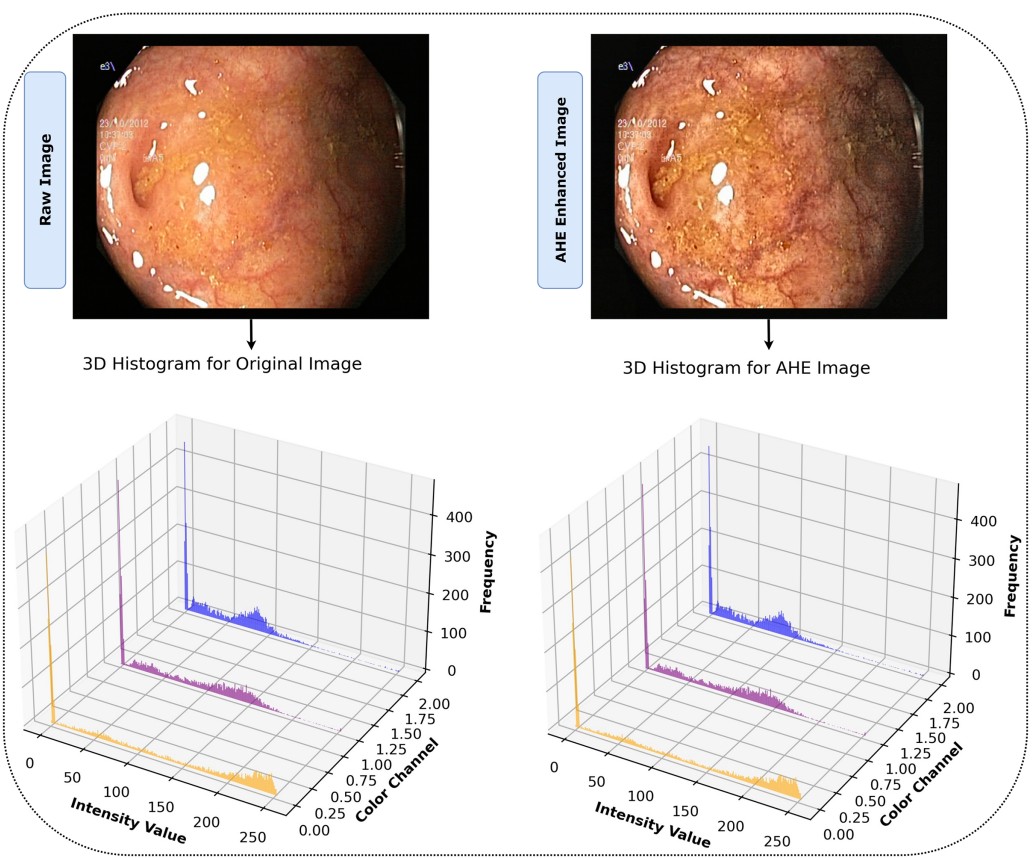

**Figure 4** **After AHE processing, images display heightened contrast and clarity, revealing detailed features for superior interpretation *vs*. their originals.**

ability to analyze medical images. The research used well-established models, including EfficientNetB0 (*Wu et al., 2020*; *Nigam et al., 2021*), MobileNetV2 (*Sandler et al., 2018*; *Buiu, Dănăilă & Răduţă, 2020*), ResNet50V2 (*Santos-Bustos, Nguyen & Espitia, 2022*; *Praveen et al., 2022*), DenseNet121 (*Li et al., 2020*), VGG16 (*Great Learning, 2022*), and Xception (*Chollet, 2016*), as a component of our technique.

1) EfficientNetB0: Originating from Google, this model is acclaimed for its exceptional scalability and adeptly accommodating various image dimensions. The MBConv block, enhanced with a squeeze-and-excitation component, functions as an inverted residual block, streamlining performance without amplifying the number of parameters. Such efficiencies render it an optimal selection for applications in medical imaging (*Buiu, Dănăilă & Răduţă, 2020*; *Santos-Bustos, Nguyen & Espitia, 2022*; *Praveen et al., 2022*).

2) MobileNetV2: The architecture of MobileNet is based on using a pointwise convolution approach. This approach speeds up the procedure while lowering computing requirements. Its improved version, V2, streamlines the design further, enhancing efficiency (*Great Learning, 2022*; *Chollet, 2016*).

3) ResNet50V2: ResNet50V2 incorporates a revised inter-block deep residual network structure designed to flow information between blocks seamlessly for improved model

---

**Algorithm 1** Adaptive histogram equalization (AHE) for image enhancement

1:    **procedure** AdaptiveHistEqualize (Image $I$, ClipLimit $C$, TileGridSize $T$)

2:      $I_{YCbCr} \leftarrow$ Convert $I$ from RGB to YCbCr color space

3:      Extract $Y$ channel as $I_Y$

4:      Divide $I_Y$ into non-overlapping tiles of size $T$

5:      **for** each tile in $I_Y$ **do**

6:         Initialize histogram array $H [0 \dots \mathrm{max\_intensity}]$ to zeros

7:         Initialize CDF array $CDF [0 \dots \mathrm{max\_intensity}]$ to zeros

8:         **for** each pixel $p$ in the tile **do**

9:            $H [\text{intensity of } p] \leftarrow H [\text{intensity of } p] + 1$

10:         **end for**

11:         $CDF[0] \leftarrow H[0]$

12:         **for** 1 *to* max_intensity **do**

13:            $CDF[i] \leftarrow CDF[i-1] + H[i]$

14:         **end for**

15:         Normalize $CDF$ using $C$

16:         **for** each pixel $p$ in the tile **do**

17:            $p_{\text{new intensity}} \leftarrow normalized\ CDF\ [\text{intensity of } p]$

18:         **end for**

19:      **end for**

20:      Apply bilinear interpolation between adjacent tiles in $I_Y$

21:      Replace $Y$ channel in $I_{YCbCr}$ with the equalized $I_Y$

22:      Convert $I_{YCbCr}$ back to RGB color space

23:      **return** Enhanced color image $I$

24:    **end procedure**

---

accuracy. The main idea behind residual blocks, referred to as "skip connections," emphasizes the resilience and effectiveness of the CNN architecture (*Das, Santosh & Pal, 2020*; *He et al., 2016*; *Akter et al., 2021*).

4) DenseNet121: The DenseNet-121 model has 121 layers and is part of the DenseNet series. Its classification subnetwork includes the $7 \times 7$ global average pooling and the 1000D fully connected layer (*Akter et al., 2021*).

5) VGG16: VGG16 is a deep convolutional neural network model introduced by the Visual Geometry Group (VGG) from the University of Oxford. It comprises 16 layers, including 13 convolutional layers and three fully connected layers. VGG16 is known for its simplicity, utilizing only $3 \times 3$ convolutional layers stacked on top of each other in increasing depth, and has proved to be extremely effective in image recognition tasks (*Minaee et al., 2020*).

6) Xception: Xception was designed to outperform other architectures by exploiting depthwise separable convolutions, which replace standard convolutions. The primary insight behind Xception is that the cross-channel and spatial correlations in the feature maps of convolutional neural networks can be mapped separately, leading to improved performance and efficiency. It has been notably successful in various image classification tasks (*Chollet, 2016*).

The selection of EfficientNetB0, MobileNetV2, ResNet50V2, DenseNet121, VGG16, and Xception for the proposed Multi-Fusion Convolutional Neural Network (MF-CNN) model was based on a systematic strategy that highlighted the architectural variety and showed effectiveness in image processing tasks. The selection of each model and the components (AuxFL, αDO, and FuRB) was based on its distinct advantages:

- EfficientNetB0 was selected due to its optimal balance of accuracy and efficiency. Moreover, its scalable architecture enables flexible model sizing, rendering it a highly suitable candidate for the wide variety of image resolutions that are commonly encountered in endoscopy data.
- MobileNetV2 was selected based on its lightweight architecture and exceptional efficiency, which are essential for real-time analytic applications. The depthwise separable convolutions provide an ideal balance between computational workload and prediction accuracy.
- ResNet50V2 addresses the issue of vanishing gradient, enabling the training of far deeper neural networks. The MF-CNN incorporates it to guarantee strong feature extraction skills, which are crucial for capturing the intricate patterns that are characteristic of GI diseases.
- DenseNet121 was chosen based on its densely linked convolutional networks, which enable efficient feature reuse across the network. This architecture is particularly effective for learning complex attributes from endoscopic images. Its inclusion enhances the model's sensitivity to subtle features indicative of GI diseases.
- VGG16 offers very reliable feature extraction capabilities. The consecutive convolutional layers of the model effectively capture a diverse range of image features, including basic textures and complex patterns. This contributes to the MF-CNN's capacity to do thorough and comprehensive analysis.
- Xception model incorporates depthwise separable convolutions, which enhance computing efficiency without compromising accuracy. The layout of the system is especially advantageous for handling the diverse and sophisticated images seen in GI diagnostic procedures, providing an optimal combination of performance and resource efficiency.
- Auxiliary Fusing Layers (AuxFL), Fusion Residual Block (FuRB), and Alpha Dropouts (αDO) are specifically developed to improve the integration process, enhance regularization, and facilitate effective feature fusion. They handle prevalent issues such as overfitting and the optimal combination of features from multiple sources, which are essential for attaining high diagnostic accuracy.

The diversified combination ensures an extensive selection of learning features, which include efficient computation for precise recognition of patterns. A balanced combination of efficiency, accuracy, and complexity led to choosing these methods to ensure that the fusion is robust and effective across various applications. This strategic selection is in contrast to selecting alternative deep learning models, which might not provide the same range of complimentary features and demonstrated performance in various scenarios, a critical consideration in aiming for a model that is both high-performing and broadly applicable in real-world scenarios.

## Strategy for fusion model and tuning

The models outlined previously in the paper have individually exhibited daunting performance metrics in the domain of medical image diagnosis. In our study, we delved into the model fusion domain to extract even more complex and subtle characteristics from the data, capitalizing on the distinct efficiency of each model.

The underlying principle of the proposed MF-CNN is simple yet powerful, as it exploits the individual strengths of each model, creating a more all-inclusive and resilient diagnostic tool. Integrating multiple CNN models is an approach to incorporate knowledge obtained from diverse domains. The implied collaborative approach has the potential to unveil a wider array of features from medical imaging, potentially increasing prediction accuracy. The integration of each chosen model contributes systematically to the extensive framework of the MF-CNN, specifically designed for GI diagnosis. It is important to emphasize that our proposed model excels in mere integration; it involves careful consideration and circumspect modifications, such as the inclusion of previously recognized FuRB, Partial Layer Freezing and AuxFL. These incorporations are consciously executed to ensure that the final MF-CNN achieves optimal performance. However, the process of combining several models with varied architectures and approaches is not devoid of difficulties. A significant concern emerges when a model becomes overly familiar with the training data too well, including its noise and outliers, making it perform less efficiently on unseen data. The vulnerability becomes more prominent when combining models, especially on limited datasets. In their endeavor to capture every minute detail, the models might just " memorize" the training data, undermining their generalizability to new, unseen data (*Olson, Wyner & Berk, 2018*).

Recognizing this potential drawback, our study incorporated a strategic countermeasure. We integrated the residual learning strategy, a signature methodology from the ResNet model, into our fusion approach. This gave origin to the FuRB. Residual learning primarily involves using "shortcut" or "skip" connections that bypass one or more layers. This approach helps mitigate the vanishing gradient issue, ensuring that the model can still learn well even when more layers are added, particularly in the context of our fusion. Figure 5 depicts the complicated fusing process in a visual manner. It shows how each selected model contributes to the overall framework of the MF-CNN built particularly for gastrointestinal (GI) diagnosis step by step. The illustration highlights the integration of several models and the meticulous planning and thoughtful modifications, such as the

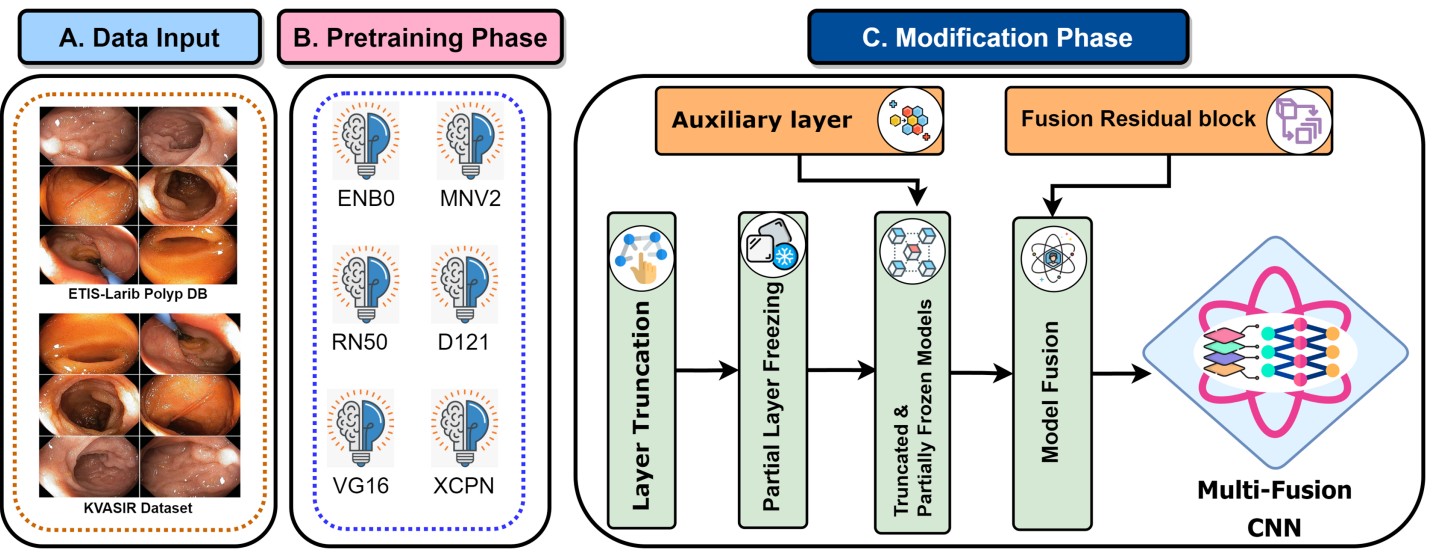

**Figure 5 MF-CNN's fusion schematic combines model strength for GI diagnosis.** The Fusion Residual Block (FuRB) ensures enhanced feature extraction and reduces overfitting risk.

addition of FuRB, that ensure the final MF-CNN achieves optimal performance while avoiding risks such as overfitting.

The initial phase used transfer learning to empower models with ImageNet's pretrained features, increasing diagnostic precision (*Orsic et al., 2019*). Following that, model layer truncation was performed to reduce the number of trainable parameters. The layers are put into a 'freezing' process during training to confirm the integrity of these enhanced pre-trained weights. The transformation of these fine-tuned models to AuxFLs assured consistent output dimension. Furthermore, these improved models were aggregated with the FuRB, resulting in an undeviating feature fusion suitable for diagnostic applications. Our research, which combines these insights and models, is a testimony to medical imaging improvements, placing the MF-CNN as a potentially transformational tool in GI diagnosis.

## Truncation of freezing layers

A systematic refinement of each designated model was undertaken in the construction of the MF-CNN. Initially, each model's upper section (often referred to as the 'head') was detached. A strategic pruning of certain layers followed this, while a substantial number of the remaining layers were suspended during the training process. The reason for these measures was to mitigate the risk of potential parameter inflation that might occur post-fusion. An excessively complex model, particularly when trained on limited datasets, is susceptible to overfitting, compromising its predictive ability (*Das, Santosh & Pal, 2020*; *He et al., 2016*).

Table 2 provides a detailed analysis of each model's parameters—both in their original and truncated states—along with the preserved features, training status, and the specific

**Table 2 Truncation settings and parameter details.**

| Model | Initial | Truncated | Features | Status | Cut-Point layer |
|---|---|---|---|---|---|
| EfficientNetB0 | 2,949,427 | 2,912,371 | 192 | Frozen | 'block6d_add' |
| MobileNetV2 | 1,738,496 | 558,656 | 96 | Frozen | 'block12_add' |
| ResNet50V2 | 4,710,592 | 1,171,456 | 512 | Frozen | 'conv3_block3_out' |
| DenseNet121 | 6,885,504 | 6,700,992 | 128 | Frozen | 'conv5_block14' |
| VGG16 | 11,174,400 | 7,635,264 | 512 | Frozen | 'block4_conv3' |
| Xception | 18,148,288 | 9,202,432 | 728 | Frozen | 'block9_sepconv3' |

layer at which truncation occurred. It's worth noting that the original architectures of these models were designed with the intent to classify up to 1,000 distinct objects. However, the scope of this analysis was narrowed down to identifying four specific categories of GI abnormalities. This disparity in classification objectives made truncation an essential step. It became more streamlined and efficient by paring down the model's architecture. An empirical approach was adopted to determine the optimal cut points, ensuring that an extensive set of features was retained despite the pruning.

## Harmonizing fusion with auxiliary fusing layers

The truncation technique introduced a challenge of mismatched cut-point dimensions among the selected models. As a result, direct integration of these models became problematic. To overcome this challenge and attain effective fusion, AuxFLs were accumulated to ensure dimensional compatibility, providing a controlled number of trainable parameters for the fused model.

Figure 6 depicts the overall construction of the AuxFLs, which are distinguished by their simplified yet strong layout. Each AuxFL is made up of three layers that are arranged sequentially: a convolutional layer (Conv), an average pooling layer (AP), and a dropout layer (αDO). Each layer has a distinct purpose. The Conv layer, which employs a h × w convolutional filter, may extract and highlight additional information. This replaces any parameters that may have been lost during truncation and guarantees that the parameter count does not increase excessively. The AP layer then enters the image and begins pooling the retrieved features to restructure and standardize their proportions, making them suitable for fusion. The decision to use average pooling rather than maximum pooling was intended since it affirms that the resulting feature maps are more representative of the underlying data, allowing for a more unified fusion. Finally, the αDO layer is incorporated for an improved regularization.

Table 3 itemizes the specific configurations and specifications of each AuxFL tailored to the requirements of the truncated variants. This detail ensures that every model's unique features are preserved and harmonized before the fusion process.

## Optimized feature fusion and FuRB implementation

AuxFLs integration ensured that all models achieved suitable cut-points, paving the road for effective feature fusion (*Salau & Jain, 2019*; *Dhiman et al., 2023*; *Liu et al., 2023*;

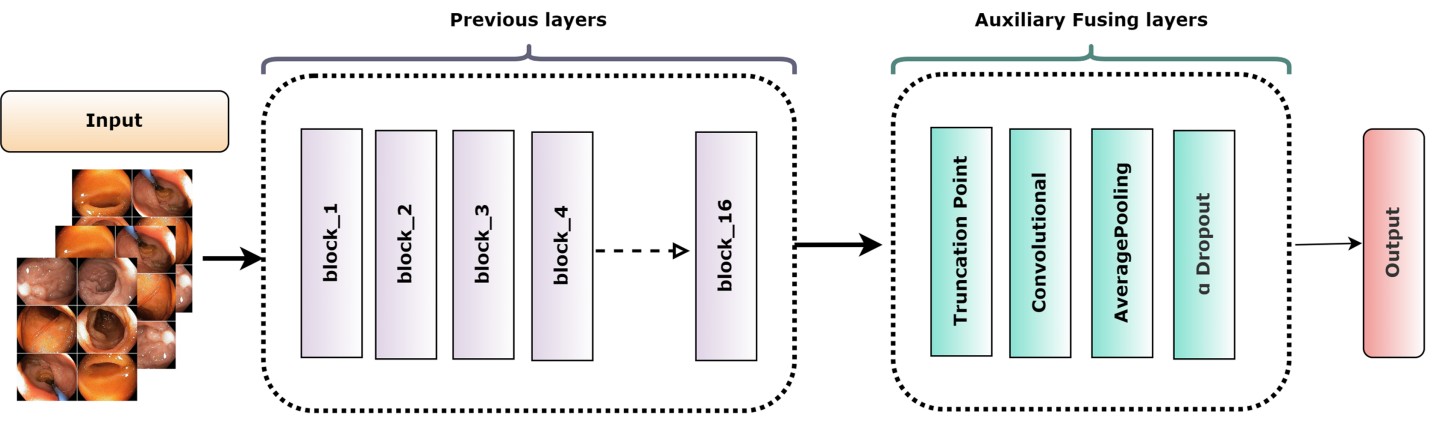

**Figure 6 The structure of the auxiliary fusing layer.**

**Table 3 Specifications of the auxiliary layers for each truncated model.**

| Model | Shape | Conv | AP | αDO |
|---|---|---|---|---|
| EfficientNetB0 | 72 × 192 | f = 192; K = 1; S = 1; Padding = Valid; activation = SeLU; initializer = LeCun Norm | Pool Size = 1; S = 1; Padding = Valid | Rate = 0.2 |
| MobileNetV2 | 142 × 96 | f = 192; K = 8; S = 1; Padding = Valid; activation = SeLU; initializer = LeCun Norm | Pool Size = 1; S = 1; Padding = Valid | Rate = 0.2 |
| ResNet50V2 | 282 × 512 | f = 192; K = 6; S = 1; Padding = Valid; activation = SeLU; initializer = LeCun Norm | Pool Size = 3; S = 3; Padding = Valid | Rate = 0.2 |
| DenseNet121 | 72 × 128 | f = 192; K = 1; S = 1; Padding = Valid; activation = SeLU; initializer = LeCun Norm | Pool Size = 1; S = 1; Padding = Valid | Rate = 0.2 |
| VGG16 | 282 × 512 | f = 192; K = 6; S = 1; Padding = Valid; activation = SeLU; initializer = LeCun Norm | Pool Size = 3; S = 3; Padding = Valid | Rate = 0.2 |
| Xception | 142 × 728 | f = 192; K = 8; S = 1; Padding = Valid; activation = SeLU; initializer = LeCun Norm | Pool Size = 1; S = 1; Padding = Valid | Rate = 0.2 |

*Montalbo, 2023*). On the other hand, the inherent complexity produced by this fusion triggered worries about excessive adherence to training data. This study employed a recognized integrated FuRB solution to address this concern. This FuRB, inspired by the ResNetV2 block's "full pre-activation" approach, was upgraded with a dropout layer (αDO), assuring optimized performance without increasing computing costs. Significantly,

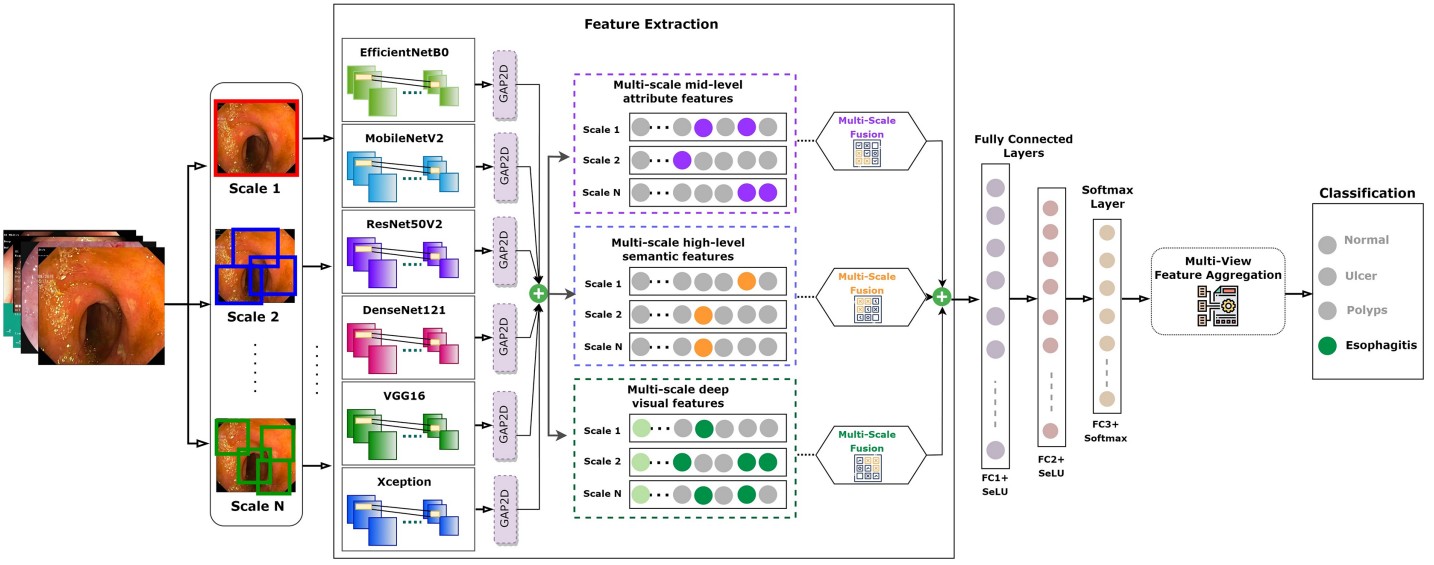

**Figure 7 Architecture of the integrated multi-fusion convolutional neural network.**

**Table 4 A comprehensive overview of selected hyperparameters and corresponding values for MF-CNN training.**

| Hyper-Parameter | Value |
| --- | --- |
| Batch size | 32 |
| Epochs | 100 |
| Optimizer | Adam |
| LR | 0.0001 |

adding the FuRB allowed the model to avoid performance saturation, improving overall efficacy. Figure 7 depicts the FuRB as well as the extensive MF-CNN framework in detail.

## MF-CNN hyperparameter optimization

The MF-CNN's hyperparameters and loss function were specified before beginning the training process. Table 4 shows the hyperparameters that were employed during training. While pre-trained networks, which are frequently trained on large datasets, excel at extracting hierarchical features, they are especially useful for smaller datasets. The majority of the models mentioned before are included in Keras. Keras-tune was first used for model fine-tuning and hyperparameter optimization, employing grid search-a prominent parameter-tuning approach. The following were the preliminary choices:

- Size of batch: 16, 24, 32, 64, 100
- Count of epochs: 30, 50, 100, 150, 250, 300
- Rate of learning: 0.0000001 to 0.1
- Optimizers: 'SGD', 'RMSprop', 'Adadelta', 'Nadam', 'Adam', 'Adamax'

The grid search approach was used to determine optimal parameter values for deep learning models. All the previously mentioned optimizers were tried for the unique MF-CNN technique. To avoid memory difficulties, the batch size for this study was fixed at 32, which was changed based on dataset size and system parameters. For its efficiency, the Adam optimizer (*Akter et al., 2021*) with a learning rate (LR) of 0.0001 was used. The model utilized Adam's adaptive capabilities to attain optimal performance in just 100 epochs, even with a low learning rate.

## MF-CNN hyperparameter and learning rate optimization

This study employed the ReduceLROnPlateau callback to ensure optimal performance without surpassing optimum accuracy levels. This is particularly beneficial if a model's LR becomes too aggressive during training. If there is no improvement in accuracy over two consecutive epochs, the ReduceLROnPlateau function automatically reduces the LR by 0.5 (50%), updating it with the new LR as per Eq. (5).

$$LRnew = LR \times 0.5. \tag{5}$$

A deep learning (DL) model's performance is measured not only by its accuracy but also by its error rates. The Categorical Crossentropy Loss function, as shown in Eq. (6), was used in this study. This function minimizes differences between expected and actual classes, providing a loss measure. It is especially useful in circumstances with several classes since, unlike its binary version, it delivers a probability for simply the relevant classes (*GitHub, 2020*).

$$CCE_{loss} = -\sum_{c=1}^{m} Y_{o,c} \log(P_{o,c}). \tag{6}$$

## Ablation analysis

A commonly employed technique for fine-tuning parameters is the implementation of the grid search method. The aforementioned approach is utilized to determine parameters including the optimizer, learning rate, loss function, batch size, and dropout. Furthermore, an ablation study was performed to validate the robustness of the proposed MFCNN model. The ablation study has impacted the following elements: model combination, layers, optimizer, learning rate, and adaptive histogram equalization (AHE).

### Case 1: altering model combination

Several model combinations have been examined to determine the optimal performance of the fusion model. EfficientNetB0, MobileNetV2, ResNet50V2, DenseNet121, VGG16, and Xception are the pre-trained models that are incorporated into the proposed model. As six additional fusion model combinations were produced to execute the shifting strategy, the proposed model was combined with EfficientNetB1, MobileNet, ResNet50V2, NASNetMobile, InceptionV3, and EfficientNetB5. From Table 5, the proposed model exhibited superior performance among the seven combinations considered, achieving a validation accuracy of 98.60% and a test accuracy of 99.25%. Additionally, it experienced

**Table 5 Analyzing ablation by the alteration of multi-fusion model combinations.**

| Model combination | Validation accuracy | Validation loss | Test accuracy | Test loss | Performance |
|---|---|---|---|---|---|
| EfficientNetB0<br>MobileNetV2<br>ResNet50V2<br>DenseNet121<br>VGG16<br>Xception | 98.60% | 13.60% | 99.25% | 6.74% | Identical |
| EfficientNetB0<br>MobileNetV2<br>ResNet50V2<br>NASNetMobile<br>InceptionV3<br>Xception | 97.35% | 20.52% | 98.37% | 12.04% | Accuracy reduced |
| EfficientNetB0<br>MobileNetV2<br>ResNet50V2<br>NASNetMobile<br>InceptionV3<br>EfficientNetB1 | 98.29% | 17.82% | 98.75% | 8.11% | Accuracy reduced |
| EfficientNetB0<br>MobileNetV2<br>ResNet50<br>DenseNet121<br>VGG16<br>Xception | 95.20% | 49.63% | 97% | 30.41% | Accuracy reduced |
| EfficientNetB0<br>MobileNetV2<br>ResNet50<br>DenseNet121<br>InceptionV3<br>Xception | 98.25% | 11.04% | 98.62% | 10.34% | Accuracy reduced |
| EfficientNetB1<br>MobileNetV2<br>ResNet50<br>NASNetMobile<br>InceptionV3<br>Xception | 95.45% | 46.39% | 97% | 26.90% | Accuracy reduced |
| EfficientNetB1<br>MobileNet<br>ResNet50<br>NASNetMobile<br>VGG16<br>EfficientNetB5 | 87.55% | 59.66% | 88.62% | 55.80% | Accuracy reduced |

the least validation loss at 13.60% and test loss at 6.74%. A slight reduction in accuracy (98.29%) and loss (11.82%) in both validation and test when compared to the proposed model was observed in an alternative model combination consisting of the initial three MFCNN models in addition to NasNetMobile, InceptionV3, and EfficientNetB1. An additional model combination, which included the initial three MFCNN models in addition to NasNetMobile, InceptionV3, and EfficientNetB1, exhibited a marginal decline

**Table 6 Ablation study examination by changing the GlobalAveragePooling2D (GAP2D) layer.**

| Applied layers | Validation accuracy | Validation loss | Test accuracy | Test loss | performance |
|---|---|---|---|---|---|
| GlobalAveragePooling2D | 98.60% | 13.60% | 99.25% | 6.74% | Identical |
| AveragePooling2D | 95.80% | 42.52% | 97.37% | 29.08% | Accuracy reduced |
| GlobalMaxPooling2D | 97.37% | 16.71% | 97.29% | 10.48% | Accuracy reduced |
| MaxPooling2D | 97.20% | 18.19% | 97.62% | 14.79% | Accuracy reduced |

in accuracy for both the validation and test tasks (98.29% and 98.75%), respectively, in comparison to the proposed model. Furthermore, it faced an increased loss of 17.82% and 8.11% for validation and testing, respectively.

### Case 2: altering global average pooling 2D layer

A Global Average Pooling2D (GAP2D) layer has been added to traditional convolutional neural networks (CNNs) to find a link between feature maps and their corresponding categories. Specifically, Global Average Pooling 2D is characterized by its ability to reduce network size and prevent overfitting. The layer undergoes a systematic replacement process to evaluate the effects of AveragePooling2D, GlobalMaxPooling2D, and MaxPooling2D on the network's performance. The accuracy of GlobalMaxPooling2D is nearly identical to that of GlobalAveragePooling2D, as illustrated in Table 6.

### Case 3: altering optimizers and learning rates

To determine the optimal optimizer and learning rate combination, learning rates of 0.0001, 0.001, and 0.01 were applied with each optimizer, including Adam, Nadam, SGD, RMSprop, and Adagrad. This method required testing each optimizer at these three separate learning rates in a sequential way. The objective was to identify which pair of optimizers and learning rate achieved the best performance. Based on the findings presented in Table 7, Adam resulted as the most robust combination, attaining a validation accuracy of 98.6% and a test accuracy of 99.25%, with a learning rate of 0.0001. Furthermore, this combination produced the least amount of test loss, which was documented at 6.74%. This finding highlights Adam's efficacy within the specified context and learning rate.

### Case 4: impact of adaptive histogram equalization in MF-CNN

Table 8 is a vital aspect in illustrating the actual impact of adaptive histogram equalization (AHE) on the performance of our model. By comparing the model's metrics in the presence and absence of AHE, this comparison provides clear verification for the contribution of AHE.

The model that incorporates AHE processing achieves superior accuracy in validation and testing (98.60% and 99.25%, respectively) in comparison to the model that does not utilize AHE (97.00% and 97.50%). This significant enhancement highlights the contribution of AHE in improving the model's ability to generalize and its precision in handling unseen data. The efficacy of AHE is demonstrated by metrics including precision,

**Table 7 Altering optimizer and learning rate (LR) to examine ablation study.**

| Optimizers | Learning rate | Validation accuracy | Validation loss | Test accuracy | Test loss | Performance |
|---|---|---|---|---|---|---|
| Adam | 0.0001 | 98.6% | 13.6% | 99.25% | 6.74% | Identical |
| | 0.001 | 92.44% | 65.18% | 92% | 54.41% | Accuracy reduced |
| | 0.01 | 85.75% | 61.79% | 87.25% | 43.83% | Accuracy reduced |
| Nadam | 0.0001 | 97.25% | 12.55% | 98.19% | 7.65% | Accuracy reduced |
| | 0.001 | 97.39% | 20.33% | 97.62% | 16.52% | Accuracy reduced |
| | 0.01 | 94.4% | 50.65% | 96.24% | 30.54% | Accuracy reduced |
| SGD | 0.0001 | 93.3% | 18.59% | 93.5% | 15.42% | Accuracy reduced |
| | 0.001 | 96.35% | 15.14% | 97.75% | 11.82% | Accuracy reduced |
| | 0.01 | 97.64% | 11.66% | 99% | 6.42% | Accuracy reduced |
| RMSprop | 0.0001 | 96.45% | 38.1% | 97.75% | 24.89% | Accuracy reduced |
| | 0.001 | 95.74% | 41.88% | 95.74% | 40.06% | Accuracy reduced |
| | 0.01 | 82.84% | 50.02% | 83.24% | 17.45% | Accuracy reduced |
| Adagrad | 0.0001 | 96.05% | 14.45% | 96.62% | 12.32% | Accuracy reduced |
| | 0.001 | 97.94% | 8.63% | 98.5% | 5.51% | Accuracy reduced |
| | 0.01 | 95.39% | 38.88% | 96.74% | 24.57% | Accuracy reduced |

**Table 8 Comparing impact on proposed model performance using AHE.**

| Processing method | Validation accuracy | Test accuracy | Precision | Recall | F1-Score |
|---|---|---|---|---|---|
| With AHE | 98.60% | 99.25% | 99.27% | 99.25% | 99.25% |
| Without AHE | 97.00% | 97.50% | 97.75% | 97.50% | 97.60% |

recall, and F1-score, which go beyond accuracy. The model that utilized AHE obtained higher precision, recall, and F1-score values (99.27%, 99.25%, and 99.25%, respectively) than the model that did not incorporate AHE (97.75% precision, 97.50% recall, and 97.60% F1-score). These enhancements emphasize the effectiveness of AHE in optimizing the model's capacity to correctly classify data points and maintaining a balance between precision and recall.

# COMPARATIVE ANALYSIS OF ALL APPLIED MODELS

## Experimental setup

To develop and test our MF-CNN, we used a computing system with an AMD Ryzen 7 3800X CPU, 32 GB of RAM, and an NVIDIA GeForce RTX 2080 Ti GPU. The hardware arrangement was selected for its superior computing power and efficiency in handling extensive datasets and performing complex deep-learning computations. The use of the NVIDIA GeForce RTX 2080 Ti GPU, loaded with its superior CUDA cores and huge memory bandwidth, facilitated the efficient training of our model, resulting in a reduction in the time needed for both model training and assessment. This configuration represents a

**Table 9 The cost and complexity analysis of applied deep learning models.**

| Model | Parameters | Operations (FLOPs) | Complexity description | Inference time (ms/image) | Size in MB |
|---|---|---|---|---|---|
| EfficientNetB0 | $5.3 \times 10^6$ | $3.9 \times 10^8$ | Efficiency focused with compound Scaling | <10 | 21.2 |
| MobileNetV2 | $3.5 \times 10^6$ | $3.0 \times 10^8$ | Lightweight with depthwise separable convolutions | <10 | 14 |
| ResNet50V2 | $25.6 \times 10^6$ | $4.0 \times 10^9$ | Deep with residual connections | 10–30 | 102.4 |
| DenseNet121 | $8 \times 10^6$ | $2.88 \times 10^9$ | Densely connected layers for feature propagation | 10–30 | 32 |
| VGG16 | $138 \times 10^6$ | $15.5 \times 10^9$ | Deep with many fully connected layers | 20–50 | 552 |
| Xception | $22.9 \times 10^6$ | $8.4 \times 10^9$ | Depthwise separable convolutions for efficiency | 10–30 | 91.6 |
| Proposed Model (MF-CNN) | $150 \times 10^6$ | $20 \times 10^9$ | Optimized with fusion of models with reduced layers and efficient architectures | 20–80 | 580 |

widely accessible yet effective computing environment that achieves a balance between accessibility and performance, thereby assuring the replication and applicability of our research across diverse contexts.

## Cost and complexity analysis

Table 9 thoroughly analyzes various deep learning models compared to the proposed optimized MF-CNN, highlighting significant factors, including inference efficiency, computational demands, and model complexity.

The MF-CNN, with $150 \times 10^6$ parameters and $20 \times 10^9$ floating point operations (FLOPs), stands out as one of the most complex and computationally intensive models in the comparison. The network's extensive parameter count signifies its fundamental and intricate nature, enabling it to capture a wide array of critical features for achieving optimal performance in complex tasks. The MFCNN needs a significant number of computations for each inference, a requirement that not only ensures thorough data processing but also requires considerable computational resources. This value is much higher than that of models such as MobileNetV2 and EfficientNetB0, renowned for their minimal computational demands and high efficiency. The MFCNN's complexity is described as an "optimized fusion of models with reduced layers and efficient architectures," indicating that a conscious decision was made during design to maintain a balance between the practical efficiency of the fusion model and its inherent complexity. The optimization above plays a critical role in effectively handling the computational cost and model's size (580 MB), rendering it a robust yet resource-intensive solution.

The MFCNN has a greater size and more FLOPs than other models, such as DenseNet121, VGG16, and Xception, indicating a higher resource consumption. Still, part of what makes it function so well is precisely this intricacy. While this may be more than for more lightweight models, the MF-CNN's projected inference time of 20–80 ms per image shows its broad processing capabilities, which are essential for attaining cutting-edge outcomes in demanding applications. Although the proposed MFCNN model demands the most resources among the models, its improved performance capabilities justify its demands. This solution maintains a balance between intensive computational

requirements and sophisticated features, rendering it well-suited for situations in which optimal performance is essential and adequate computational resources are accessible.

## RESULT AND ANALYSIS FINDINGS

The present section provides an in-depth evaluation of the proposed Multi-Fusion Convolutional Neural Network (MF-CNN) using established deep learning (DL) benchmarks. To assess the MF-CNN's efficacy, its results are compared with top-tier DL models. For this analysis, we utilize an image dataset segmented into three categories: training, validation, and test sets. Diagnostic performance is determined through a comprehensive array of methods, including a confusion matrix, precision-recall (P-R) curve, and the receiver operating characteristic (ROC) curve, with particular emphasis on the area under the curve (AUC) metric. Furthermore, the chapter highlights the MF-CNN's distinctiveness by comparing its performance to other significant studies in the field.

### Evaluation of diagnostic performance metrics

The conventional methods depend primarily on the anticipated elements of the confusion matrix, *i.e.*, true positives (TP), false positives (FP), true negatives (TN), and false negatives (FN), derived from the validation and test datasets. These components are used to evaluate the overall diagnostic accuracy of vision-based models in the field of medical imaging. The suggested model's overall accuracy, precision, recall, and F1-score are evaluated using the following Eqs. (7) to (10) based on the values of the elements of the confusion matrix (*Minaee et al., 2020*).

$$Accuracy = \frac{TP + TN}{TP + TN + FP + FN} \tag{7}$$

$$Precision = \frac{TP}{TP + FP} \tag{8}$$

$$Recall/Sesitivity = \frac{TP}{TP + FN} \tag{9}$$

$$f_1 - score = \frac{2 \times Precision \times Recall}{Precision + Recall}. \tag{10}$$

### Diagnostic performance analysis

The dataset in this study was divided into training, validation, and test sets, allowing for a more detailed performance comparison with other cutting-edge approaches. The confusion matrices for both the validation and test datasets are shown in Figs. 8 and 9, demonstrating the predictive accuracy of the proposed MF-CNN model. As shown in Fig. 8, the model made just 28 mistakes out of 2,000 samples in the validation set, attaining a remarkable accuracy rate of 98.6% and an error rate of 1.4%. Similarly, in Fig. 9, the model misclassified only six of the 800 samples in the test set, resulting in an accuracy of 99.25% and an error rate of 0.75%.

The findings in Table 10 indicate the performance measures of the MF-CNN model, such as accuracy, precision, recall, and F1-score, as determined from the three-way split
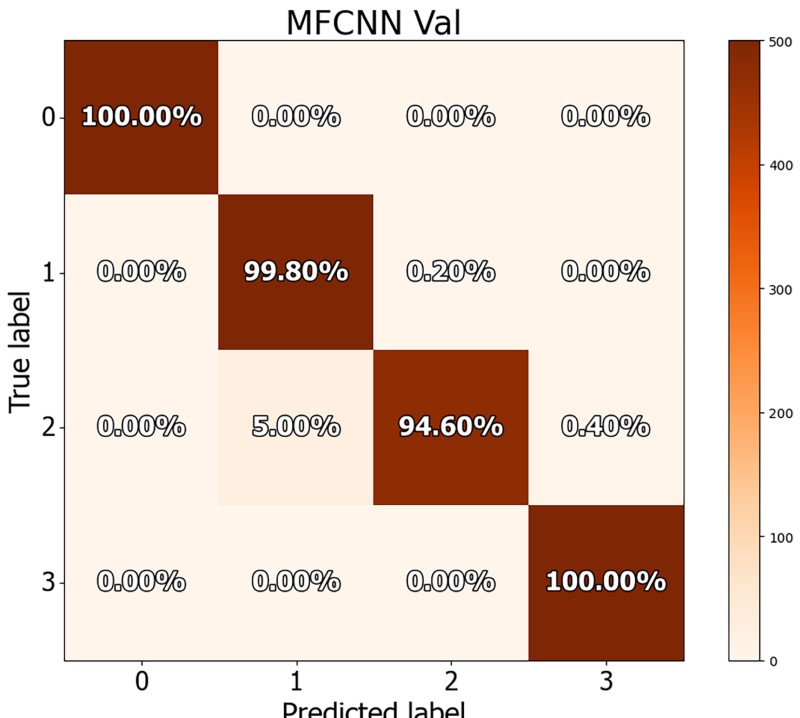

**Figure 8 Confusion matrices for MF-CNN on validation datasets.** Its showcase the model's accuracy by detailing correct and incorrect predictions in each set.

approach. These values were obtained by analyzing the data in the associated confusion matrix. The MF-CNN model attained an impressive accuracy of 98.60% on the validation set and 99.25% on the test set. Furthermore, the model displayed exceptional consistency across additional criteria. Precision, recall, and F1-score for the validation set were 98.65%, 98.6%, and 98.6%, respectively. In comparison, the test set reported precision, recall, and F1-score values of 99.27%, 99.25%, and 99.25%, respectively.

Tables 11 and 12 provide an in-depth evaluation of the model's diagnostic accuracy for each distinct instance. The MF-CNN's proficiency was slightly lower when diagnosing ulcer and polyp diseases compared to the normal and esophagitis situations. As shown in Table 11, among the validation data, polyp diagnosis had the lowest accuracy at 94.60%, whereas ulcer diagnoses had a higher rate of 99.80%. When the model's accuracy for the polyps diagnosis was evaluated using the test data from Table 12, it was recorded at 97.00%. Such findings imply that the MF-CNN's capacity to distinguish polyp cases is considerably limited when compared to other situations. Despite this minor variance, the MF-CNN provided an excellent overall diagnostic performance.

## ROC curve analysis

The receiver operating characteristics (ROC) curve was used by the MF-CNN model to explore its diagnostic capabilities more deeply, analyzing sensitivity and specificity across multiple thresholds (*Nour & Polat, 2020*). As Eq. (11) specified, the model's sensitivity examined its accuracy in identifying instances with mucosal abnormalities. Similarly, as

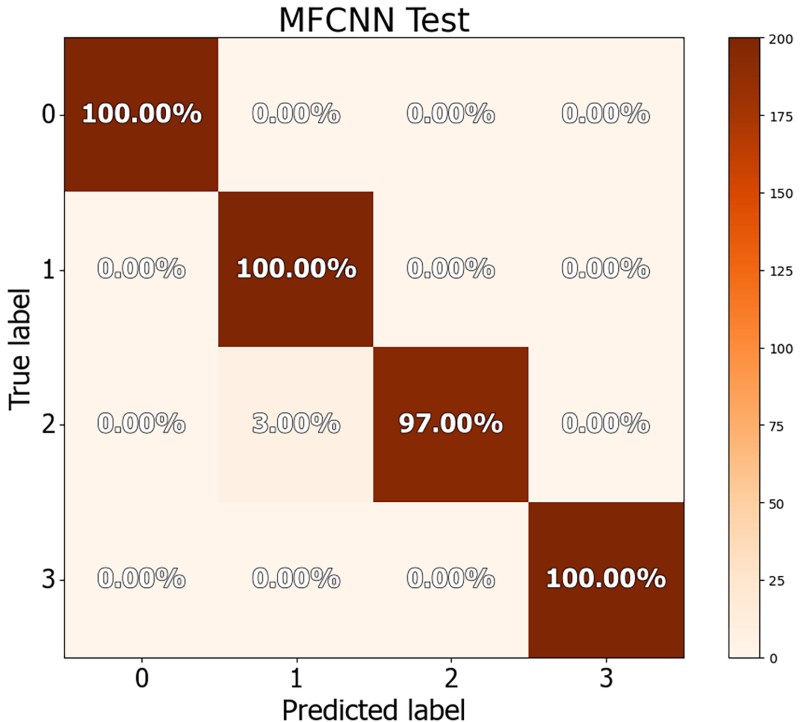

**Figure 9 Confusion matrices for MF-CNN on test datasets.** Its showcase the model's accuracy by detailing correct and incorrect predictions in each set.

**Table 10 The diagnostic performance of the proposed MF-CNN model.**

| Dataset | Accuracy | Precision | Recall | F1-Score | Samples |
|---|---|---|---|---|---|
| Validation | 98.60% | 98.65% | 98.60% | 98.60% | 2,000 |
| Test | 99.25% | 99.27% | 99.25% | 99.25% | 800 |

**Table 11 Diagnostic accuracy of MF-CNN for specific cases using the validation dataset.**

| Case | Accuracy | Precision | Recall | F1-Score |
|---|---|---|---|---|
| Normal | 100% | 1.00 | 1.00 | 1.00 |
| Ulcer | 99.80% | 0.95 | 0.99 | 0.97 |
| Polyps | 94.60% | 0.99 | 0.94 | 0.97 |
| Esophagitis | 100% | 0.99 | 1.00 | 0.99 |

defined in Eq. (12), specificity assessed the model's accuracy in identifying situations where the mucosa showed no abnormalities.

$$Sensitivity = \frac{Instances\ of\ correctly\ diagnosed\ samples\ with\ abnormalities}{Total\ number\ of\ samples\ with\ abnormalities} \qquad (11)$$

**Table 12 Detailed analysis of MF-CNN diagnostic accuracy for specific cases using the test dataset.**

| Case | Accuracy | Precision | Recall | F1-Score |
|---|---|---|---|---|
| Normal | 100% | 1.00 | 1.00 | 1.00 |
| Ulcer | 100% | 0.97 | 1.00 | 0.98 |
| Polyps | 97.00% | 1.00 | 0.97 | 0.98 |
| Esophagitis | 100% | 1.00 | 1.00 | 1.00 |

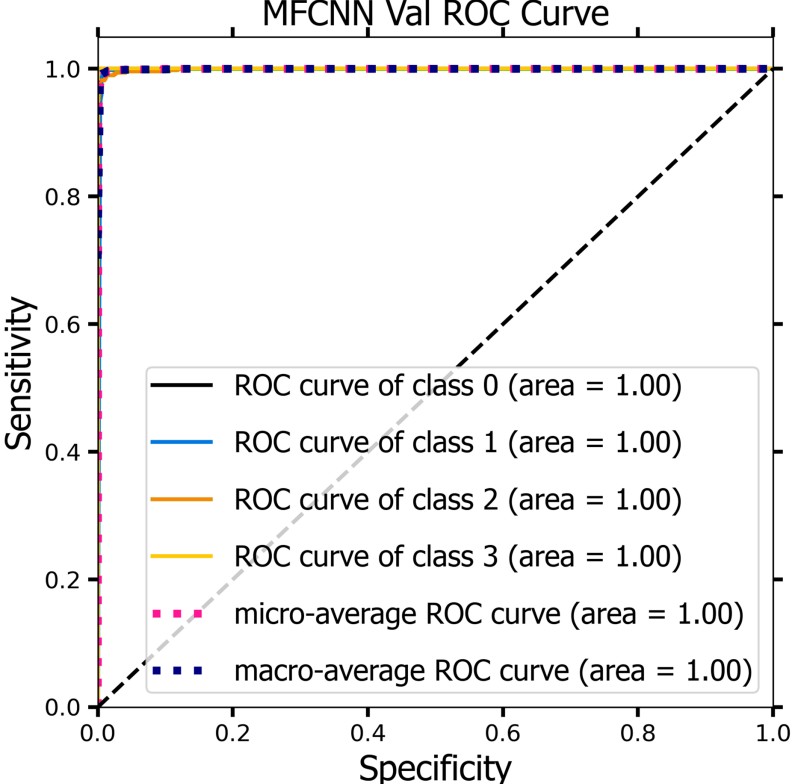

**Figure 10 ROC curve of the MF-CNN model derived from the validation datasets, illustrating consistent diagnostic accuracy with an average AUC of 1.00.**

$$Specificity = \frac{Instances\ of\ correctly\ diagnosed\ samples\ without\ abnormalities}{Total\ number\ of\ samples\ without\ abnormalities}. \tag{12}$$

Figures 10 and 11 demonstrate the ROC curve for the MF-CNN model, derived from the sensitivity and specificity metrics of the validation and test datasets. The AUC values averaged 1.00 for both the validation set represented in Fig. 10 and the test set in Fig. 11. This highlights the model's consistent diagnostic precision across disparate thresholds. The findings confirm the model's outstanding performance on both datasets (*Jeni, Cohn & De La Torre, 2013*).

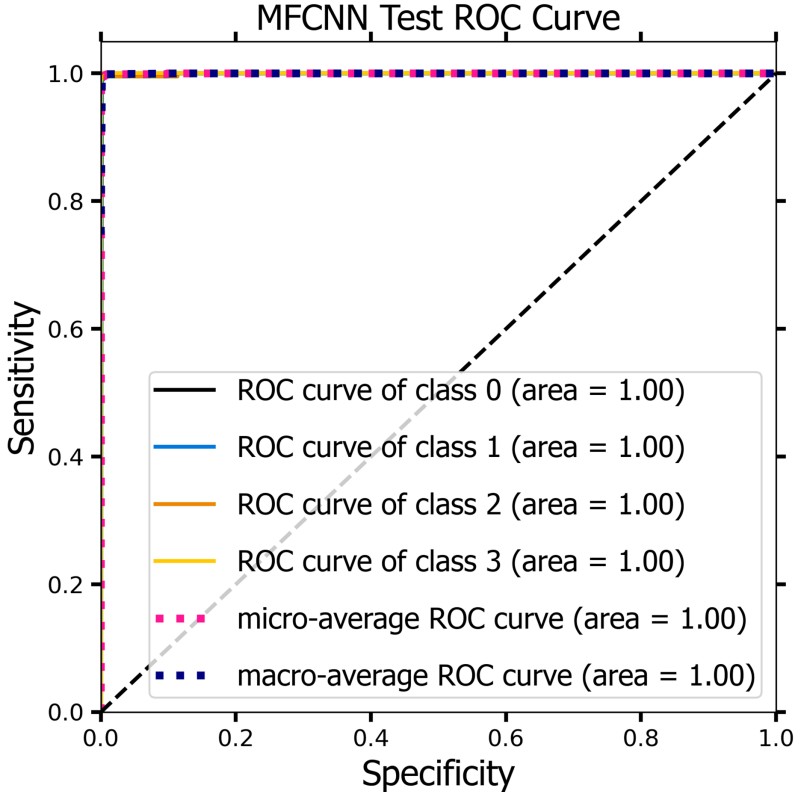

**Figure 11** The ROC curve of the MF-CNN model derived from the test datasets, illustrating consistent diagnostic accuracy with an average AUC of 1.00.

## PR curve analysis

Similar to the ROC curve, the Precision-Recall (P-R) curve is formulated based on the recall or sensitivity metric. This curve precisely balances precision, frequently referred to as the positive predictive value, and recall. Its ability to adjust for varying thresholds grants the P-R curve a unique edge in providing a holistic assessment of the model's efficacy with the dataset (*Pizer et al., 1987*). Figures 12 and 13 vividly portray the P-R curve of the advanced MF-CNN model, distilled from insights collected from both validation and test datasets. Observations from the validation set, represented in Fig. 12, revealed a micro-average P-R curve value of 0.996. In comparison, the test set, illustrated in Fig. 13, reported a marginally superior value of 0.998. Such findings accentuate the model's prowess in rapidly diagnosing an array of GI diseases. An essential conclusion from this analysis is the model's unwavering performance consistency. This is evident from the nearly uniform P-R curve across both datasets, suggesting minimal anomalies. Such consistent outcomes reinforce the model's reliability, simultaneously mitigating any identified weaknesses.

## Comparative analysis

Figures 14 and 15 provide a detailed visualization of the accuracy and loss metrics associated with the proposed MF-CNN model. This model's robust performance is evident from its consistently increasing accuracy and decreasing loss as training progresses

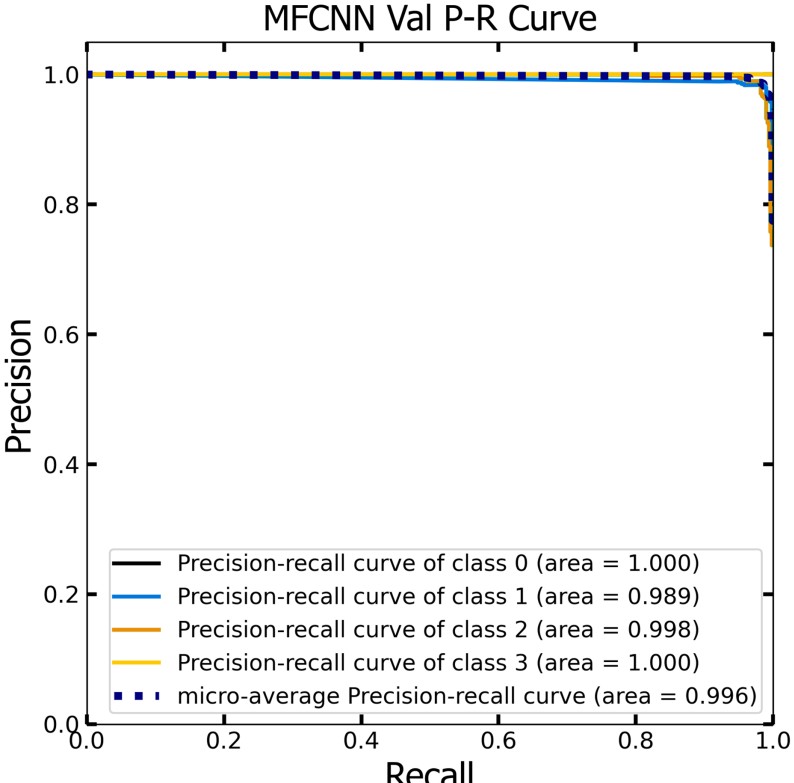

**Figure 12** The Precision-Recall (P–R) curve of validation dataset with a P–R value of 0.996 for the MF-CNN model.

through successive epochs. Figure 14, in particular, highlights the training and validation accuracy statistics. The model's progression indicates a consistent improvement in accuracy, reflecting its effective learning capability. On the other hand, the primary objective during the model's training phase is to minimize the loss. This scalar measure quantifies the difference between the model's predictions and the actual labels. A lower loss score implies more accurate predictions. This relationship is depicted in Fig. 15, which compares training loss against validation loss. Both accuracy and loss metrics are displayed against the epoch count. At the onset, during the initial epochs, the model presents its most fundamental performance, characterized by the lowest accuracy and the steepest loss. However, as training persists, both metrics exhibit significant improvement. By the 90th epoch, a significant milestone is achieved: validation accuracy maximizes, and the validation loss is at its lowest concurrently. This convergence signifies that the model's pinnacle of performance is attained at the 90th epoch, marked by peak accuracy and minimized loss.

To assess the MF-CNN's ability to diagnose certain gastrointestinal disorders, it is essential to compare its performance to various top DCNN models. This section compares the MF-CNN to well-known DCNN models using the same dataset. *Montalbo (2022)* compared their MFuRe-CNN against 11 of the best Keras pre-trained models. This work is consistent with Montalbo's comparison, as it draws on his performance assessments of
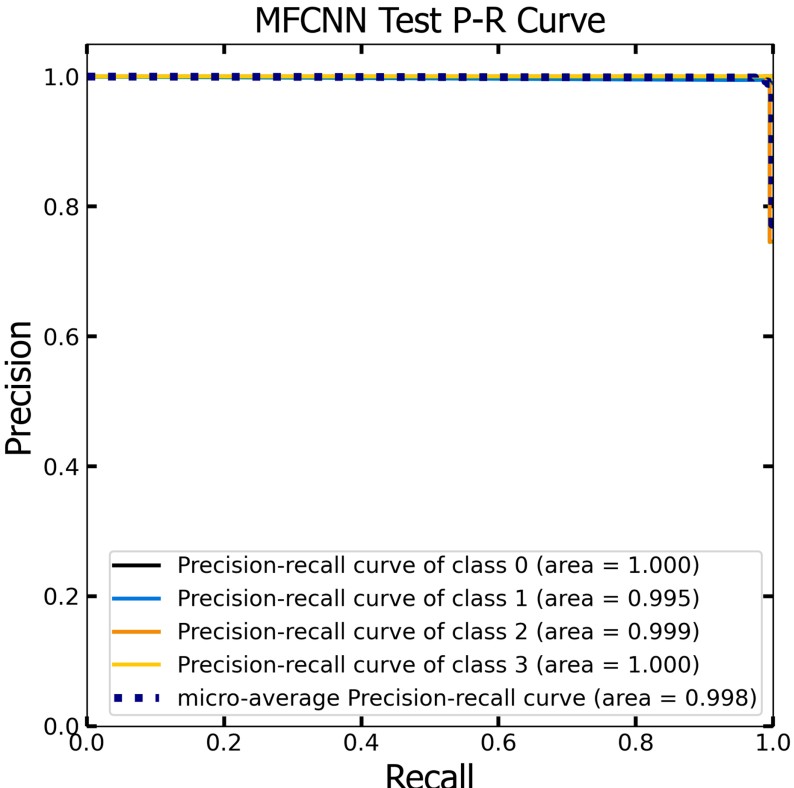

**Figure 13** The Precision-Recall (P–R) curve of test dataset with a P–R value of 0.998 for the MF-CNN model.

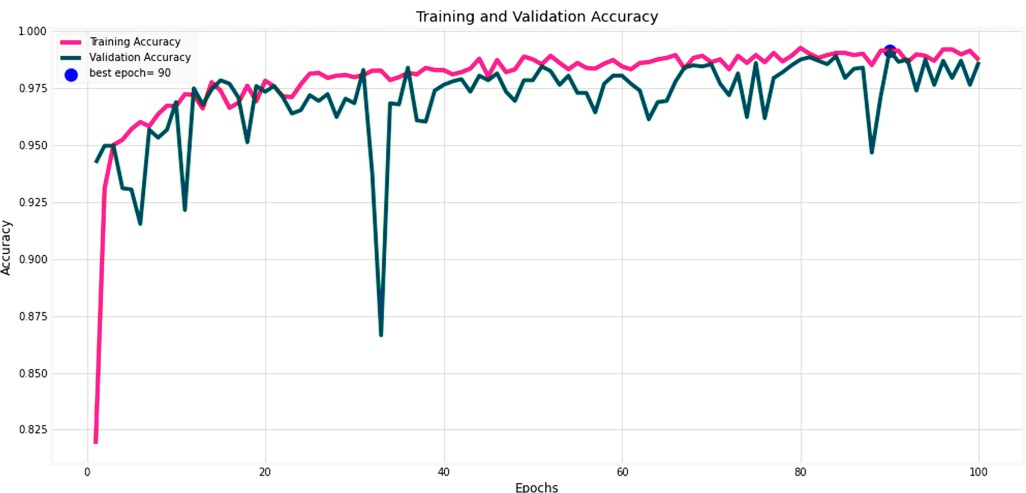

**Figure 14** Performance metrics of the MF-CNN model. Evolution of training and validation accuracy, demonstrating consistent improvement with peak accuracy attained near the conclusion of training.

these DCNNs given the common dataset. Table 13 shows that the MF-CNN scored highest with a validation accuracy of 98.60%. In comparison, the highest-ranking DCNN, MobileNetV2, attained an accuracy of 90.90%. The MF-CNN achieves a considerable

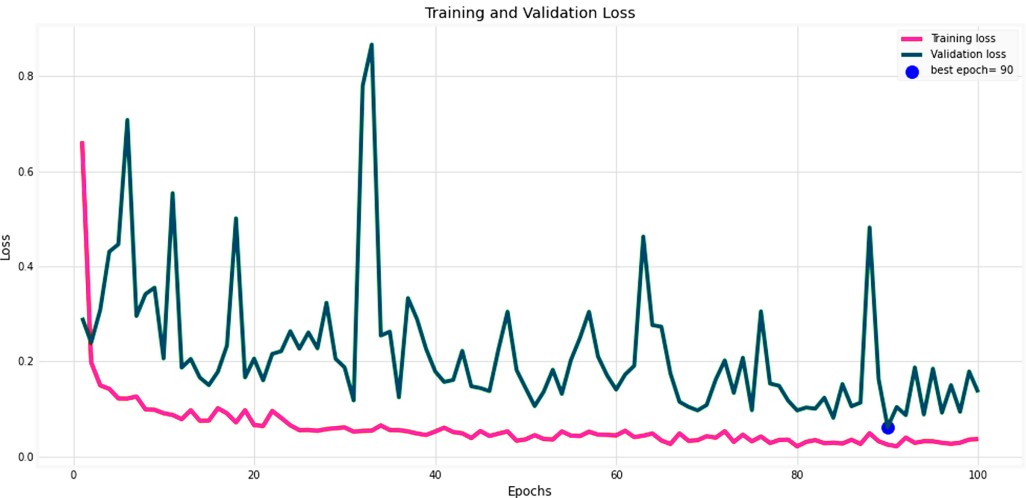

**Figure 15 Performance metrics of the MF-CNN Model.** Corresponding trends in training and validation loss, indicating the most refined model performance with minimized loss also towards the conclusion of training.

**Table 13 Validation performance comparison of the proposed MF-CNN model with leading DCNN models.**

| Model | Validation | | | | Test | | | |
|---|---|---|---|---|---|---|---|---|
| | Accuracy | Precision | Recall | F1-Score | Accuracy | Precision | Recall | F1-Score |
| MF-CNN | 98.60% | 98.65% | 98.60% | 98.60% | 99.25% | 99.27% | 99.25% | 99.25% |
| MFuRe-CNN | 96.65% | 96.67% | 96.65% | 96.65% | 97.75% | 97.75% | 97.75% | 97.75% |
| MobileNetV2 | 90.90% | 91.31% | 90.90% | 90.93% | 92.25% | 92.75% | 92.25% | 92.28% |
| EfficientNetB0 | 86.85% | 87.81% | 86.85% | 86.97% | 88.25% | 89.03% | 88.25% | 88.38% |
| InceptionV3 | 83.35% | 85.26% | 83.35% | 83.44% | 85.37% | 86.73% | 85.38% | 85.45% |
| DenseNet121 | 82.25% | 86.64% | 82.25% | 81.90% | 84.50% | 87.81% | 84.50% | 84.35% |
| Xception | 82.75% | 85.91% | 82.75% | 82.78% | 84.50% | 86.46% | 84.50% | 84.58% |
| ResNet50V2 | 83.70% | 86.72% | 83.70% | 83.81% | 84.25% | 87.20% | 84.25% | 84.42% |
| InceptionResNetV2 | 77.15% | 81.76% | 77.15% | 76.21% | 80.13% | 84.14% | 80.13% | 79.44% |
| NASNetLarge | 77.25% | 81.75% | 77.25% | 77.54% | 77.62% | 82.12% | 77.62% | 78.07% |
| VGG16 | 73.40% | 74.33% | 73.40% | 72.94% | 75.13% | 75.53% | 75.12% | 74.69% |
| NASNetMobile | 69.35% | 75.15% | 69.35% | 66.61% | 72.12% | 77.81% | 72.13% | 69.60% |
| VGG19 | 65.35% | 71.62% | 65.35% | 63.51% | 68.00% | 74.15% | 68.00% | 65.68% |

performance improvement of 7.7%. While the MF-CNN outperforms traditionally trained DCNNs, MobileNetV2 has advantages from the proposed fusion strategy. Furthermore, our MF-CNN model outperformed Montalbo's MFuRe-CNN by 1.95%, with a validation accuracy of 96.65%. Similarly, Table 13 also shows, MF-CNN outperformed MobileNetV2 with a 99.25% accuracy rating from the test dataset, where MobileNetV2 had 92.25%. An improvement of 7% showcases the supremacy of MF-CNN on this dataset. Even if we look at the customized MFuRe-CNN model, it had an accuracy of 97.75%. Compared to Montalbo's model, MF-CNN had an accuracy rate of 1.5% higher. This further illustrates

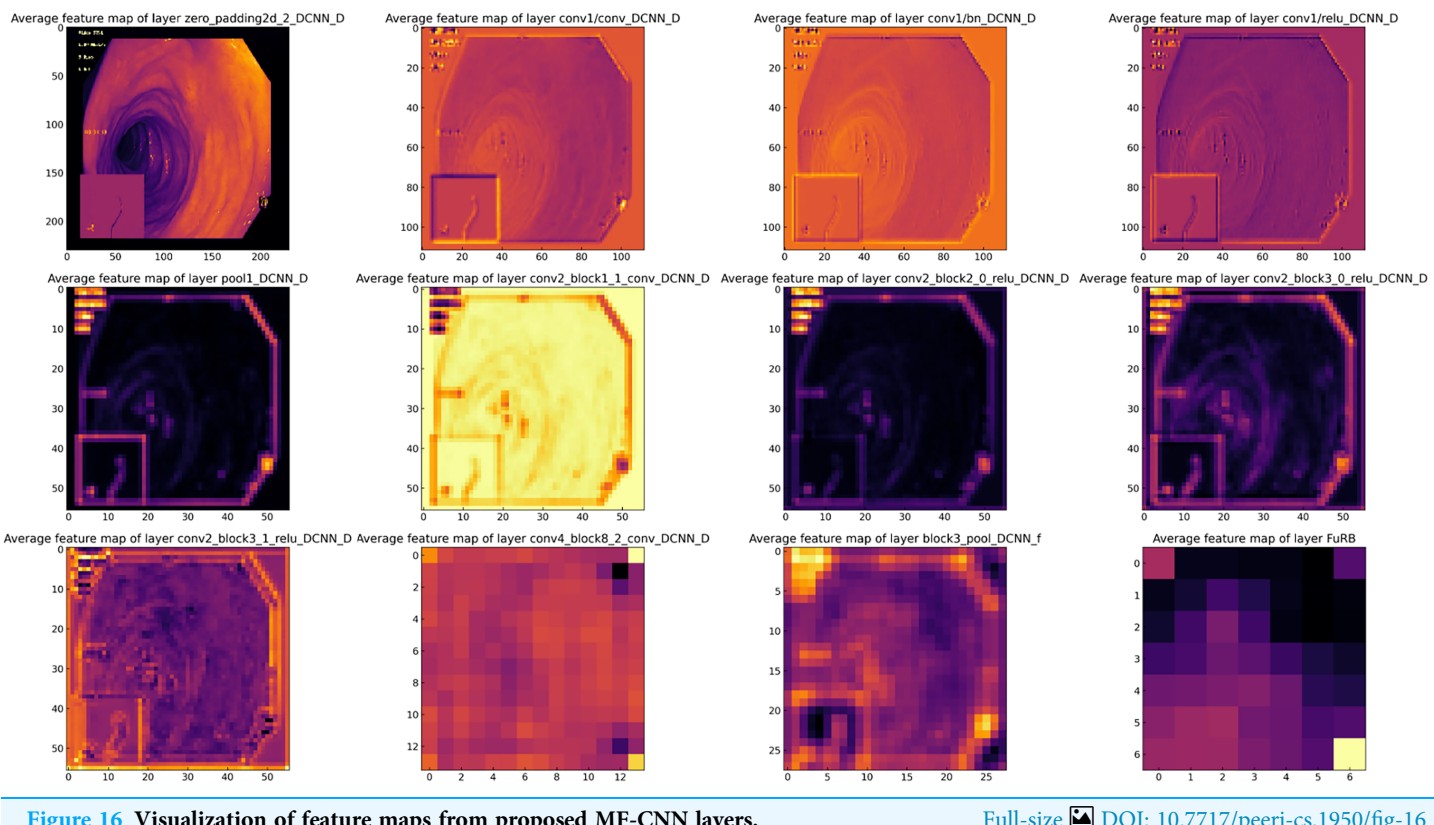

**Figure 16 Visualization of feature maps from proposed MF-CNN layers.**

## Feature map analysis of MF-CNN

Feature map analysis involves interpreting and understanding the outputs (feature maps) produced by the convolutional layers in a CNN. These feature maps depict the activations of the neurons in the layer, signifying the features in the input data (such as an image) that the neurons have detected. Figure 16 presents a visual representation of the feature maps generated by the layers of the MF-CNN. Due to the model's extensive depth, the figure selectively showcases only certain layers and blocks. In the earliest layers of the MF-CNN, as demonstrated in the feature maps, there is a distinct focus on the basic, low-level features of the input. These foundational elements can be compared to the basic building blocks of the image. For instance, the feature maps capture and exhibit aspects such as boundaries, textures, and distinct color regions. These foundational elements give us an overview of what the neural network first identifies and discerns in its initial layers. The early phases of interpretation are fundamental and relatively straightforward, concentrating on the primary structures and patterns in the data. A significant prominence can be seen in the upper left quadrant of Fig. 16. This particular section provides a visualization of the feature maps from the initial layers of the MF-CNN. The patterns

**Table 14 Comparative analysis of the proposed model's diagnostic accuracy with models from previous studies in the detection of GI diseases.**

| Authors | Methods | Accuracy |
| --- | --- | --- |
| *Montalbo (2022)* | Fusion CNN | 97.75% |
| *Fan et al. (2018)* | AlexNet | 95.16% |
| *Majid et al. (2020)* | VGG16, KNN | 96.5% |
| *Poudel et al. (2020)* | CNN, Dynamic Dilated Conv | 95.7% |
| *Hmoud Al-Adhaileh et al. (2021)* | AlexNet, GoogleNet, ResNet-50 | 97% |
| *Khan et al. (2020)* | VGG16, Transfer learning, Feature fusion | 98.4% |
| *Öztürk & Özkaya (2021)* | ResNet50, LSTM | 98.05% |
| *Olson, Wyner & Berk (2018)* | Compressed and Modified Fusion CNN | 97.75% |
| *Orsic et al. (2019)* | Swin-transformer | 87.22% |
| *Salau & Jain (2019)* | EfficientNetB0 | 98% |
| *Dhiman et al. (2023)* | InceptionV3, ResNet50 and DenseNet201 | 95% |
| *Hossain et al. (2022)* | Proposed model | 99.25% |

emerging here graphically highlight the network's ability to detect fundamental visual elements. We can observe distinct indications of contours and individual color regions, signifying the network's focus on these rudimentary components.

## Comprehensive assessment of state-of-the-art methods

In the following section, we systematically compare the efficacy of various CNN models in diagnosing gastrointestinal (GI) diseases, including our proposed MF-CNN. These models are compared comprehensively in Table 14, with an emphasis on their diagnostic accuracy. Upon analysis of the data, it becomes evident that the proposed method consistently outperforms other models in its domain. The outstanding performance metrics observed in this framework can be attributed to its advanced design and cutting-edge development methodologies. The table serves as a visible depiction of the advances made in the current study and supports the claim that our proposed method has established a new standard. We have demonstrated our model's exceptional accuracy through testing and evaluation, making it a leading tool for detecting GI diseases.

## LIMITATIONS OF THE STUDY

The research on the MF-CNN for identifying gastrointestinal disorders is comprehensive, although it has limitations that do not significantly affect its overall findings and contributions. The dataset we used came from KVASIR and ETIS-Larib Polyp DB. These databases do not cover all gastrointestinal disorders in the world, but they give us a good starting point for building and testing our model. The integration of six sophisticated DCNNs challenges the MF-CNN computational efficiency, but this does not affect the model's demonstrated accuracy. While the current scope of the MF-CNN is limited to four major gastrointestinal tract abnormalities, this focus does not undermine its potential applicability to a boarder range of conditions in future developments. Furthermore,

positive findings have already been seen in the experimental settings, however, real-world clinical validation is still required. Ultimately, the possibility of making more algorithmic advancements indicates continuous opportunities for strengthening the model rather than being a drawback of the present research. These constraints are relatively minor and provide opportunities for further investigation rather than having a large influence on the existing accomplishments of the study.

## CONCLUSION AND RECOMMENDATION

In this study, the developed MF-CNN marks a significant advancement in the field of GI tract abnormality detection, outperforming traditional state-of-the-art DCNNs. By integrating the strengths of six DCNN models (EfficientNetB0, MobilenetV2, ResNet50V2, DenseNet121, VGG16, and Xception) into a unified pipeline and refining them with streamlined layers, selective freezing, AuxFLs, and αDOs, the MF-CNN achieves exceptional diagnostic accuracy scores of 98.60% and 99.25% on validation and test datasets, respectively. This approach not only surpasses the constraints of existing methodologies in terms of accuracy and adaptability but also establishes a new standard in the analysis of GI medical images. To further improve the performance of the MF-CNN, future directions involve enhancing its robustness across diverse datasets, optimizing computational efficiency, and exploring algorithmic or structural enhancements. During our study, we experienced several challenges, particularly in addressing the diversity of medical imaging and the technical requirements of handling extensive datasets. The presence of variations in image quality and the need for significant data preparation emphasize the importance of developing more flexible and efficient data handling systems. Moving forward, we recommend focusing on improving the MF-CNN's resilience over a wider range of datasets. This may include making more algorithmic improvements or using cutting-edge data augmentation methods. Furthermore, it is still important to maximize the computing efficiency of the model, either by exploring compact model structures or more sophisticated optimization techniques. These endeavors will enhance the MF-CNN's practical applicability and expand its effectiveness to include a wider array of medical imaging tasks.

### Funding
The authors received no funding for this work.

### Competing Interests
The authors declare that they have no competing interests.

### Author Contributions
- Tanzim Hossain conceived and designed the experiments, performed the experiments, analyzed the data, performed the computation work, prepared figures and/or tables, authored or reviewed drafts of the article, and approved the final draft.

- F M Javed Mehedi Shamrat conceived and designed the experiments, performed the experiments, analyzed the data, performed the computation work, prepared figures and/ or tables, authored or reviewed drafts of the article, and approved the final draft.
- Xujuan Zhou conceived and designed the experiments, analyzed the data, prepared figures and/or tables, authored or reviewed drafts of the article, and approved the final draft.
- Imran Mahmud analyzed the data, prepared figures and/or tables, authored or reviewed drafts of the article, and approved the final draft.
- Md. Sakib Ali Mazumder analyzed the data, prepared figures and/or tables, authored or reviewed drafts of the article, and approved the final draft.
- Sharmin Sharmin analyzed the data, prepared figures and/or tables, authored or reviewed drafts of the article, and approved the final draft.
- Raj Gururajan analyzed the data, prepared figures and/or tables, authored or reviewed drafts of the article, and approved the final draft.

### Data Availability

The code is available in the Supplemental File.

The WCE Curated Colon Disease Dataset (which contains the KVASIR Dataset and ETIS-Larib-Polyp DB Dataset) is available at Kaggle: https://www.kaggle.com/datasets/ francismon/curated-colon-dataset-for-deep-learning?fbclid=IwAR3X7ZtLL6otehwzh RdIgayTW3BuCQ8MleNKjXU6EBfuiImAybIE320vBmE.

### Supplemental Information

Supplemental information for this article can be found online at http://dx.doi.org/10.7717/ peerj-cs.1950#supplemental-information.

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
