# Peer review of "Development of a multi-fusion convolutional neural network (MF-CNN) for enhanced gastrointestinal disease diagnosis in endoscopy image analysis"

_PeerJ Computer Science, doi:10.7717/peerj-cs.1950_

## Round 0.1 · original submission · Major Revisions

Based on the referee reports, I recommend a major revision of the manuscript. The author should improve the manuscript, taking carefully into account the comments of the reviewers in the reports, and resubmit the paper.

Reviewers 1 & 2 have suggested that you cite specific references. You are welcome to add it/them if you believe they are relevant. However, you are not required to include these citations, and if you do not include them, this will not influence my decision.

**Language Note:** PeerJ staff have identified that the English language needs to be improved. When you prepare your next revision, please either (i) have a colleague who is proficient in English and familiar with the subject matter review your manuscript, or (ii) contact a professional editing service to review your manuscript. PeerJ can provide language editing services - you can contact us at copyediting@peerj.com for pricing (be sure to provide your manuscript number and title). – PeerJ Staff

Reviewer 1 ·

Basic reporting

1. The study appears to draw heavily from an existing body of research without offering substantial differentiation. The authors assert that "their study" introduces certain components that are already known. It would be advisable for the authors to rephrase their work to emphasize that they are "utilizing" or "incorporating" components like the Auxiliary Fusing Layers, Fusion Residual Block, and Alpha Dropouts.
2. The integration of AuxFL, aDO, and FuRB already addresses the issue of overfitting, a topic that has been explored in prior studies. The authors should consider rephrasing their findings to highlight how they have integrated these existing components to produce novel results.
3. More comprehensive details and emphasis are needed for Image Enhancement, specifically Adaptive Histogram Equalization (AHE), given its potential impact on the model's overall performance. It is crucial to provide mathematical proof of how AHE is applied in the study and quantify the impact it has on the original images.
4. The authors should thoroughly review and cite the paper "Fusing compressed deep ConvNets with a self-normalizing residual block and alpha dropout for cost-efficient classification and diagnosis of gastrointestinal tract diseases" (doi: https://doi.org/10.1016/j.mex.2022.101925), which presents the use of the components mentioned in the paper.

5. The study would benefit from an ablation study to elucidate the significance of each component in the proposed MF-CNN. Additionally, the ablation study should reveal how the new model outperforms the previous MFuRe-CNN.
6. A more comprehensive range of experiments should be considered. It is essential to explore whether the observed improvements are primarily due to preprocessing and AHE. The authors should provide a detailed explanation and empirical results to support their claims.
7. The overall structure of the paper requires improvement to enhance clarity and readability. Thorough proofreading is necessary to rectify errors and inconsistencies.
8. The authors should provide additional details on how they selected the models to fuse. A transparent and well-justified approach is essential.
9. Utilizing a combination theory to determine the best model combinations is recommended. The results from these combinations should also be subjected to ablation analysis.
10. The study should address the cost and complexity of the proposed model. A discussion of the resources and computational requirements is essential for a comprehensive understanding.
11. The study lacks a practical assessment of its solution in comparison to existing methods. The authors should introduce a dedicated section that examines the practicality of their approach and employ a scientific method to do so.
12. The limitations of the study are currently unclear and inadequately presented. The authors should consider incorporating a section outlining the limitations of their research for a more comprehensive evaluation.

Experimental design

The authors may consider lookin into my comments.

Validity of the findings

The authors may consider lookin into my comments.

Additional comments

The authors may consider lookin into my comments.

Reviewer 2 ·

Basic reporting

1. The abstract is not detailed enough. Readers expect to see more detail of the methodology, results, and conclusion in the abstract. The abstract need to be greatly improved. The abstract does not show that the authors achieved much as there is no numerical justification to back the author’s claims or results of comparative analysis to show superior performance.
2. In the introduction, the authors should explain why they did it (motivation) discussing the possible outcome. Readers are primarily interested in the motivation and outcome of your research. Therefore, a good introduction should contain:
a. What is the problem to be solved?
b. Are there any existing solutions?
c. Which is the best?
d. What is the main limitation of the best and existing approaches?
e. What do you hope to change or propose to make it better?
f. How is the paper structured?
3. Please clearly highlight how your work advances the field from the present state of knowledge and you should provide a clear justification for your work which should be stated at the end of literature review/ related works. The impact or advancement of the work can also appear in the conclusion.
4. The authors mentioned feature extraction but have not presented this stage in their work. A block diagram or flowchart of the steps would be helpful. The authors should look for recent works on feature extraction to cite. An example of such recent literature which the authors can consult amongst others is:
-Feature Extraction: A Survey of the Types, Techniques, Applications, 5th IEEE International Conference on Signal Processing and Communication (ICSC), Noida, India, pp. 158-164 (2019). DOI: 10.1109/ICSC45622.2019.8938371

In addition, before feature extraction is performed you will first select the most important features which is called feature selection. Authors should consult papers in this area also.
5. Related works section is not sufficient. The authors should improve on this section as they have left many papers out. Normally, it’s the gaps in work of others that the authors are expected to fill. Therefore, at the end of your review section state the problems in this field with appropriate reference and tell readers which one your work addresses.
The authors should consult and cite:
-Detection and classification of gastrointestinal disease using convolutional neural network and SVM. Cogent Engineering, Vol. 9(1), 2084878, pp. 1-24, 2022. DOI: 10.1080/23311916.2022.2084878

6. The computing systems specifications, e.g. RAM size and processing capacity was not mentioned? This should be mentioned to guide those who want to replicate the work.
7. The major contributions of the paper should be listed after the literature review, after stating the limitations of other studies. You would then state which of the challenges your work addresses.
8. What is the principle of operation of the proposed multi-fusion convolutional neural network that makes it work well in this problem?

Experimental design

Details of the experimentation need to be explained better.

Validity of the findings

The results need to be compared with existing works. Most of the figures in this paper are not clear enough. The authors should endeavour to change them. The authors need to discuss the results, especially those in the Table. The reason why the proposed technique performs better has not been explained.
I was hoping to see more results and discussion as more results could be presented to make the work. The Limitations of the proposed study need to be discussed before conclusion.

---

## Round 0.2 · Major Revisions

Kindly revise the manuscript as per the reviewer suggestions and resubmit it.

Reviewer 1 ·

Basic reporting

.

Experimental design

.

Validity of the findings

.

Additional comments

It would be advisable for the authors to refrain from using the term "pioneering" in describing their solution, as it heavily relies on a pre-existing model with minor modifications. Instead, they should emphasize the foundational aspects of their approach, elucidating the specific elements that served as the basis for their work. Providing a more in-depth exploration of their motivation behind leveraging these components will contribute to a clearer understanding of how they ingeniously crafted an enhanced solution through thoughtful modifications and adaptations.

Reviewer 2 ·

Basic reporting

1. Lastly, no comparison of results was presented with other state-of-the-art methods which have used machine learning techniques
2. The authors should add details of the computing device and justify its use.
3. The authors did not state the amount of data used for experimentation? Or what type of data was used (primary or secondary?). Nowhere in this paper did the authors mention the data used to test the model and the mathematical derivations. It would be great to provide the link to the data. Although, the authors presented Table 2 which seems a bit general.
4. The authors should structure the paper into abstract, introduction, literature.
5. The authors can present the translators algorithm in tabular form. Also the computing systems specifications and capacity was not mentioned?
6. The Limitations of the proposed study need to be discussed before conclusion.
7. Some of the challenges encountered during the course of the study can be highlighted and future recommendations can be added at the end of the conclusion. Retitle conclusion as conclusion and recommendation.

Experimental design

.

Validity of the findings

The results need to be validated with proper metrics.

---

## Round 0.3 · accepted · Accept

Author has addressed the reviewers' comments properly. Thus I recommend publication of the manuscript.

Reviewer 1 ·

Basic reporting

Pass.

Experimental design

Pass.

Validity of the findings

Pass.

Additional comments

The authors have addressed my concerns and given these changes. I think I can give an "Accept"